# Two distinct conformational states define the interaction of human RAD51-ATP with single-stranded DNA

Ineke Brouwer[1,†,‡], Tommaso Moschetti[2,‡], Andrea Candelli[1], Edwige B Garcin[3], Mauro Modesti[3], Luca Pellegrini[2,*] ⓘ, Gijs JL Wuite[1,**] ⓘ & Erwin JG Peterman[1,***] ⓘ

## Abstract

An essential mechanism for repairing DNA double-strand breaks is homologous recombination (HR). One of its core catalysts is human RAD51 (hRAD51), which assembles as a helical nucleoprotein filament on single-stranded DNA, promoting DNA-strand exchange. Here, we study the interaction of hRAD51 with single-stranded DNA using a single-molecule approach. We show that ATP-bound hRAD51 filaments can exist in two different states with different contour lengths and with a free-energy difference of ~4 $k_B$T per hRAD51 monomer. Upon ATP hydrolysis, the filaments convert into a disassembly-competent ADP-bound configuration. In agreement with the single-molecule analysis, we demonstrate the presence of two distinct protomer interfaces in the crystal structure of a hRAD51-ATP filament, providing a structural basis for the two conformational states of the filament. Together, our findings provide evidence that hRAD51-ATP filaments can exist in two interconvertible conformational states, which might be functionally relevant for DNA homology recognition and strand exchange.

**Keywords** DNA repair; homologous recombination; RAD51; single-stranded DNA

**Subject Categories** DNA Replication, Repair & Recombination; Structural Biology

**The EMBO Journal (2018) 37: e98162**

## Introduction

The efficient repair of DNA damage is crucial for chromosome integrity, since it prevents mutations, chromosomal aberrations and errors in essential processes such as transcription, replication and chromosome segregation (Hoeijmakers, 2001). One of the key mechanisms for repairing DNA double-strand breaks (DSBs) is homologous recombination (HR). This is a multistep process, where after occurrence and detection of a DSB, the broken DNA ends are processed by the end-resection machinery to produce 3′ single-stranded DNA (ssDNA) overhangs (San Filippo *et al*, 2008; Holthausen *et al*, 2010). Subsequently, RAD51 is recruited to the DNA overhang to form a right-handed helical nucleoprotein filament (NPF) in an ATP-dependent manner (Bianco *et al*, 1998). The NPF is responsible for DNA sequence homology recognition in a duplex DNA, usually the sister chromatid, and formation of a joint intermediate that will serve as a priming site for DNA synthesis needed to copy the missing information (Benson *et al*, 1994). In the cell, HR is tightly regulated, for example by the tumour-suppressor protein BRCA2 (Sung & Klein, 2006), which is known to mediate the loading of hRAD51 onto RPA-coated ssDNA, making use of its ability to bind hRAD51 with its BRC-repeat domain (Wong *et al*, 1997; Chen *et al*, 1998; Carreira *et al*, 2009).

Our previous single-molecule work on the interaction of hRAD51 with ssDNA focused mainly on the assembly of the NPF (Candelli *et al*, 2014) and on the disassembly of hRAD51 from double-stranded DNA (dsDNA; van Mameren *et al*, 2009b). The latter study showed that hRAD51 disassembles through a pause–burst mechanism from NPF ends, dominated by ATP hydrolysis of the RAD51 monomers at filament ends. This process is highly dependent on the tension in the dsDNA template: disassembly stalls completely at forces above 50 pN. In addition, although the NPF can readily form in the presence of both ATP (van Mameren *et al*, 2009b) and ADP (Hilario *et al*, 2009), hRAD51 disassembly from dsDNA is critically dependent on ATP hydrolysis (van Mameren *et al*, 2009b). X-ray crystallography has identified the location of the ATP-binding pocket between adjacent monomers in the NPF (Conway *et al*, 2004). Therefore, it is likely that ATP binding and hydrolysis play a significant role in the conformation and stability

1   Department of Physics and Astronomy and LaserLaB, Vrije Universiteit Amsterdam, Amsterdam, The Netherlands
2   Department of Biochemistry, University of Cambridge, Cambridge, UK
3   Cancer Research Center of Marseille, CNRS UMR7258, Inserm U1068, Institut Paoli-Calmettes, Aix-Marseille Université, Marseille, France
    *Corresponding author. Tel: +44 1223 760469; E-mail: lp212@cam.ac.uk
    **Corresponding author. Tel: +31 205987987; E-mail: g.j.l.wuite@vu.nl
    ***Corresponding author. Tel: +31 205987576; E-mail: e.j.g.peterman@vu.nl
    ‡These authors contributed equally to this work
    †Present address: Department of Gene Regulation, The Netherlands Cancer Institute, Amsterdam, The Netherlands

of hRAD51 NPFs. Understanding the disassembly of hRAD51 from ssDNA is important as the intrinsic stability of hRAD51-ssDNA NPFs is likely to affect the reaction of strand exchange during homologous recombination (Taylor *et al*, 2015, 2016) and the ability of hRAD51 to protect ssDNA gaps present at stalled replication forks (Kolinjivadi *et al*, 2017). Yet, the mechanism of hRAD51 disassembly from ssDNA and how it is affected by factors such as ssDNA template tension and ATP hydrolysis remain unknown.

One well-established feature of the NPF structure is its remarkable conservation across the kingdoms of life, which is accompanied by considerable conformational polymorphism, resulting from variations in helical rise and twist (Ogawa *et al*, 1993; Yu *et al*, 2001; Liu *et al*, 2004). Such heterogeneity, captured by structural analysis, is likely to reflect the NPF ability to undergo conformational transitions that alter its pitch, giving rise to extended or compact filament states. These conformational states are thought to represent important yet poorly understood stages in the filament dynamics that underlie the mechanism of DNA-strand exchange.

Electron microscopy data have shown that the nature of the nucleotide bound to RAD51 has a marked effect on filament conformation: NPFs containing the structural analogue of the ATP transition state ADP·AlF4$^-$ adopt an extended conformation with a pitch of 9.9 nm, while ATPγS-bound NPFs adopt a more compact conformation with a pitch of ~7.6 nm (Yu *et al*, 2001). More recently, high-resolution cryo-electron microscopy (cryoEM) studies of the presynaptic NPF in the presence of the ATP analogue AMP-PNP have measured an average filament pitch of 10.3 nm with 6.4 protomers per turn (Short *et al*, 2016) and confirmed its structural polymorphism, with 80% of the filaments having pitch values in the range of 9.5–11.0 nm; a similar cryoEM analysis of the active form of the NPF reported a filament pitch of 10.0 nm with 6.3 protomers per turn (Xu *et al*, 2017).

These nucleotide-dependent variations in the NPF structure reflect critical differences in biochemical function: the ATP-bound extended hRAD51 filament is competent to perform strand exchange, while the ADP-bound compact hRAD51 filament is inactive and might represent an intermediate state before disassembly (Bugreev & Mazin, 2004). Thus, ATP hydrolysis affects the behaviour of the hRAD51 NPF, as a dynamic entity that is able to switch between multiple conformations (Yu *et al*, 2001). Although large-scale differences in NPF structure, such as variations in pitch, have been widely reported, local changes in filament structure at the single protomer level have not been described yet. The only exception comes from a crystallographic model of yeast Rad51 in filament form, which showed the presence of different protomer interfaces that were proposed to be functionally relevant (Conway *et al*, 2004).

Similar conformational transitions have also been observed for hRAD51 filaments on dsDNA. Experiments using magnetic tweezers (Ristic *et al*, 2005; Atwell *et al*, 2012) have shown that hRAD51 binds dsDNA in two distinct modes with different pitch and that, upon ATP hydrolysis, the NPFs can switch to a compact state before NPF disassembly. Changes in DNA twist can also trigger transitions between the two modes. Furthermore, transitions between extended and compact filament conformations have been observed using single-molecule FRET measurements of the bacterial hRAD51

orthologue RecA (Kim *et al*, 2014), and ATP hydrolysis and cooperative structural changes between adjacent RecA molecules were invoked to explain the observed NPF dynamics.

Much remains to be learned about the structure and dynamic behaviour of the different conformational states of the hRAD51-ssDNA NPF and how this variety of structural modes relates to function. In this study, we used a combination of dual-trap optical tweezers, single-molecule fluorescence microscopy and microfluidics to study the dynamic behaviour of hRAD51-ssDNA NPFs. We find that the ATP-bound NPF can exist in two interconvertible states that differ in contour length and free energy. Furthermore, we demonstrate the existence of two different protomer interfaces in the crystal structure of hRAD51-ATP filaments, in agreement with the single-molecule data. Together, our findings provide experimental evidence for the postulated existence of defined conformational states of the presynaptic hRAD51 NPF, which might be relevant to the processes of homology recognition and DNA-strand exchange during homologous recombination.

## Results

### hRAD51-ssDNA NPF disassembly

The first property of hRAD51-ssDNA NPFs that we investigated is the disassembly kinetics of hRAD51 from ssDNA. For this, we made use of our experimental set-up that has been described in detail before (van Mameren *et al*, 2008, 2009b; Gross *et al*, 2010; Candelli *et al*, 2011; Heller *et al*, 2014; Brouwer *et al*, 2016). In brief, dual-trap optical tweezers are used to capture micron-sized streptavidin-coated beads (Fig 1A); using a computer-controlled microscope stage and a microfluidics system (u-Flux, LUMICKS B.V.; Fig 1B), fast buffer exchange can be achieved such that an individual molecule of end-biotinylated dsDNA is tethered between the beads. In these experiments, we used 48.5-kb λ-phage dsDNA. By applying tensions above 80 pN, the dsDNA is converted into ssDNA by force-induced melting (van Mameren *et al*, 2009a; Gross *et al*, 2011). Subsequently, this ssDNA molecule was exposed to a buffer containing a fluorescent variant of hRAD51 (hRAD51-K313-C319S-Alexa Fluor 555, referred to as hRAD51 in this study, Appendix Fig S1A–C). Because hRAD51 disassembly is triggered by ATP hydrolysis (van Mameren *et al*, 2009b), we measured the disassembly rate of hRAD51 in the presence of Mg$^{2+}$. After incubating (for 10 s to 5 min) an ssDNA molecule in the protein channel (containing 0.18 μM hRAD51), the construct was brought back to the protein-free buffer channel where the fluorescence signal was used to confirm a high coverage of the ssDNA with hRAD51 (Fig 1C, top panel). To limit the effect of photobleaching (Appendix Fig S2A and B), fluorescence images were acquired intermittently every 30 s (Fig 1C). The corresponding kymograph is shown in Fig 1D, where a stepwise reduction in fluorescence signal of the individual filaments is shown, suggesting that disassembly of hRAD51 filaments from ssDNA, as from dsDNA (van Mameren *et al*, 2009b), occurs in bursts following ATP hydrolysis at the terminal RAD51 monomer. RAD51 disassembly from ssDNA does not appear to be sequence dependent (Appendix Fig S3A and B). The decrease in total fluorescence intensity of the RAD51-ssDNA construct as a function of time is shown in Fig 1E.

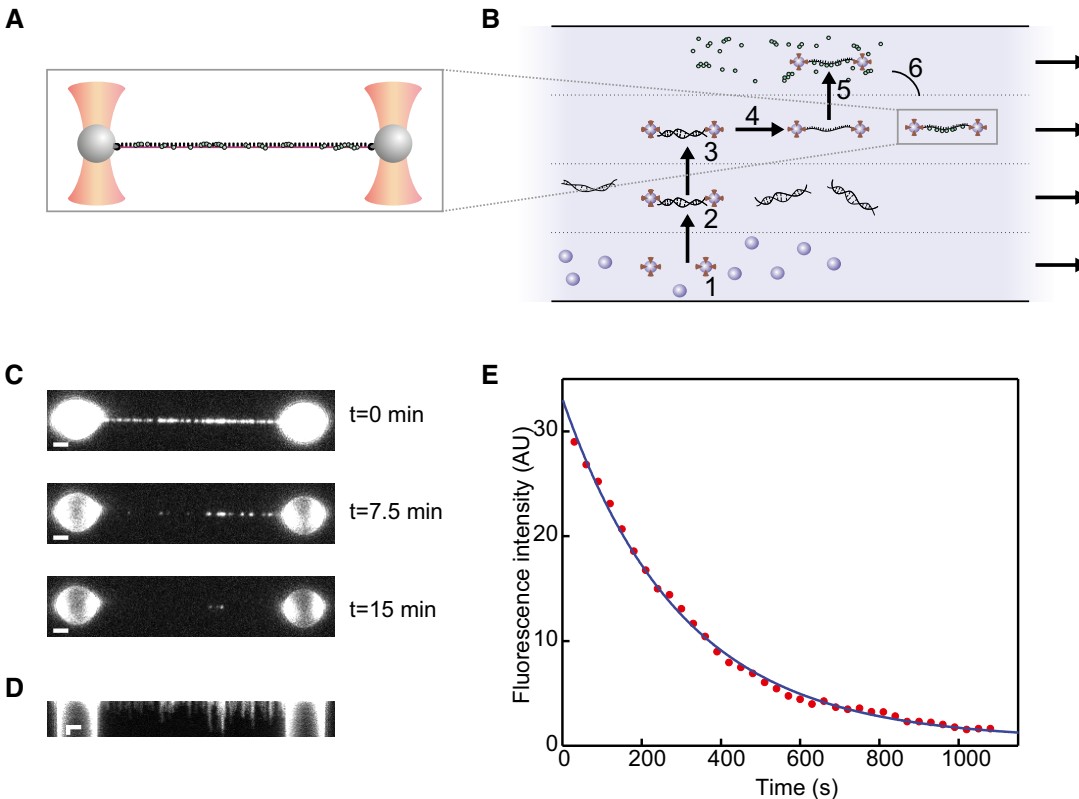

**Figure 1. Detecting fluorescent hRAD51 bound to an individual ssDNA molecule.**

A   Schematic of an ssDNA molecule (purple) tethered between two optically trapped micrometer-sized polystyrene beads (grey) with hRAD51 complexes (green) bound to the ssDNA molecule. By controlling the position of the beads, the extension of the DNA molecule can be controlled while the tension in the molecule is monitored. At the same time, the proteins can be directly visualized with single-fluorophore resolution using wide-field fluorescence microscopy.

B   The experiments are generally performed using a microfluidic flow system with four laminar channels. A typical experiment is comprised of the following steps: (1) capture of two beads; (2) tethering of a single dsDNA molecule between these beads; (3) probing the mechanical properties of the tethered dsDNA molecule, to ensure that it is a single molecule with the expected mechanical properties; (4) the tension on the dsDNA molecule is increased to generate an ssDNA molecule by force-induced melting; (5) the ssDNA is incubated in the protein channel; and (6) the hRAD51-ssDNA complex is brought into the buffer for imaging.

C   Typical fluorescence intensity snapshots of hRAD51-ssDNA complexes (buffer composition: 20 mM Tris pH 7.5, 100 mM KCl, 1 mM MgCl$_2$, 1 mM ATP, 10 mM DTT) at indicated time intervals. Scale bars: 2 μm.

D   Fluorescence kymograph of the same hRAD51-ssDNA complex as in (C). Scale bars: 2 μm (horizontal) and 5 min (vertical).

E   Integrated fluorescence intensity along the DNA of the same hRAD51-ssDNA complex as in (C and D) over time (red dataset), showing an exponential decay at a rate of $(33 \pm 1) \ 10^{-4} \ s^{-1}$, obtained from an exponential fit to the data (blue curve). After correcting for photobleaching, this gives, for this particular example, a hRAD51 disassembly rate of $(17 \pm 1) \ 10^{-4} \ s^{-1}$.

Source data are available online for this figure.

It is known that hRAD51 disassembly from dsDNA slows down at increased tension on the dsDNA template, resulting in complete stalling of disassembly at forces exceeding 50 pN (van Mameren *et al*, 2009b). We therefore tested whether similar force dependence occurs in hRAD51 disassembly from ssDNA by measuring the average disassembly times at three different forces (Fig 2A–C). The observed disassembly rates of $(14 \pm 2) \ 10^{-4} \ s^{-1}$ (at 5 pN; $N = 8$; dissociation rates are obtained from exponential fits to the fluorescence intensity traces, errors are fitting errors), $(15 \pm 1) \ 10^{-4} \ s^{-1}$ (at 20 pN; $N = 16$) and $(12 \pm 1) \ 10^{-4} \ s^{-1}$ (at 50 pN; $N = 4$) show that hRAD51 dissociation from ssDNA is not influenced by tension within the range of values tested. Interestingly, a similar effect was reported previously (Candelli *et al*, 2014) for the assembly process: both nucleation and growth of hRAD51 NPFs are highly tension-dependent for dsDNA, but independent of tension for ssDNA. This

finding can be attributed to the fact that dsDNA is more rigid and resistant to length change (King *et al*, 2013). The persistence length of ssDNA is more than two orders of magnitude smaller than the one of dsDNA and is much more compliant to protein-induced structural changes, independent of the tension in the DNA.

In addition, we analysed the relative effect of ATP and ADP on NPF disassembly rates (Fig EV1A–D) and found a slightly lower rate for ADP-bound than ATP-bound NPFs. Assuming that, as for dsDNA (van Mameren *et al*, 2009b), disassembly occurs strictly from filament ends, the observed rates depend on the number of NPFs bound. Correcting for differences in initial DNA coverage yields comparable disassembly rates of $(9 \pm 2) \ 10^{-4} \ s^{-1}$ for ADP-bound filaments and $(11 \pm 2) \ 10^{-4} \ s^{-1}$ for ATP-bound filaments (Fig EV1A–D). Finally, we show that the disassembly rate is independent of ionic strength within the tested range (0–100 mM KCl

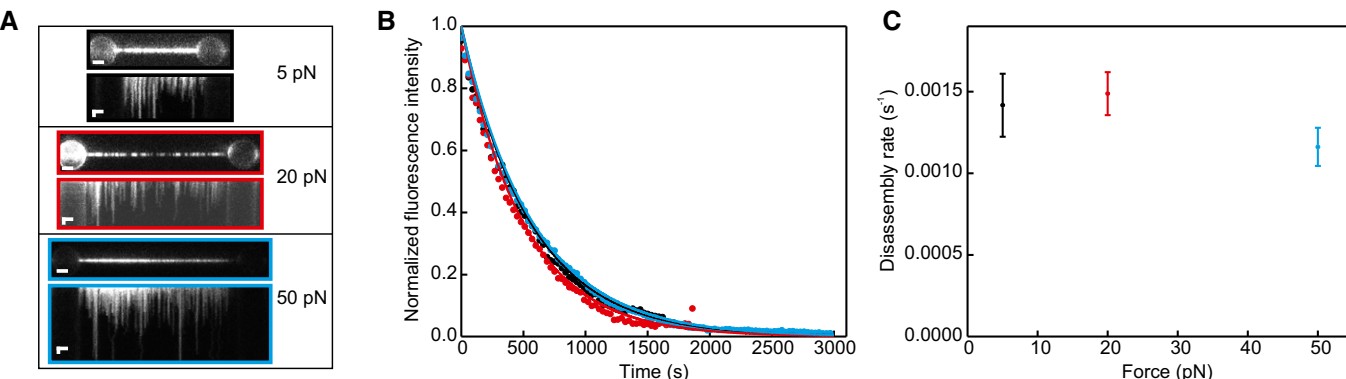

**Figure 2.  Disassembly of hRAD51 from ssDNA is independent of DNA tension.**

A   Fluorescence images and kymographs of hRAD51 disassembling from ssDNA at indicated ssDNA tensions (buffer composition: 20 mM Tris pH 7.5, 100 mM KCl, 1 mM MgCl₂, 1 mM ATP, 10 mM DTT). Images are typical examples of 16 (at 5 pN), 9 (at 20 pN) and 4 (at 50 pN) identical experiments. Scale bars: 2 µm (horizontal) and 5 min (vertical).

B   Integrated fluorescence intensity along the DNA of the same complexes as in (A) over time and corresponding an exponential fit. Fits are normalized using the amplitude and offset of the exponential fit. Coloured edges in (A) show colour of corresponding force curve.

C   Average disassembly rate as a function of tension. Error bars originate from the exponential fits on the individual datasets (number of datasets as in A).

Source data are available online for this figure.

and 1–10 mM MgCl₂; Fig EV2A–C). The combined results of these experiments indicate that disassembly from ssDNA is insensitive to tension, ionic strength and the type of nucleotide present in the buffer.

## Conformational transitions within hRAD51-ssDNA NPFs

Next, we set out to investigate the mechanical response of hRAD51-ssDNA NPFs to tension (Fig 3A–I). To this end, we incubated ssDNA molecules in a channel containing fluorescent hRAD51 until a high coverage of the DNA was achieved (2–4 min). After repositioning the hRAD51-ssDNA complex in protein-free buffer, we verified that the ssDNA was densely coated with hRAD51 using fluorescence imaging. Subsequently, we performed successive cycles of stretching and relaxation (at a rate of 0.66 ± 0.01 µm/s), while continuously monitoring the tension in the hRAD51-ssDNA complex and fluorescence intensity.

A first set of experiments was performed in a buffer containing 2 mM Ca²⁺ and 2 mM ATP. As ATP hydrolysis takes place on a timescale of 10–100 min (Bugreev & Mazin, 2004) under these conditions, the amount of hRAD51 bound to the ssDNA can be considered to remain constant. The observed force-extension and force-relaxation curves of hRAD51-ssDNA differed drastically from those of bare ssDNA (Fig 3A). In addition, we observed that the extension and relaxation curves of these complexes did not overlap: rather, the curves show significant hysteresis. Under these conditions, hRAD51 can neither bind from solution nor disassemble from the DNA, which is confirmed by the integrated fluorescence intensity and the hysteresis area, both remaining constant during the timescale of the experiments (Fig 3B and C). In addition, the minimum distance between the ssDNA ends was kept at 5 µm, to maintain the DNA in an extended conformation and to avoid interactions between filaments. Therefore, we propose that this hysteretic behaviour is caused by a force-induced transition between ATP states of the hRAD51 NPFs, which results in reversible changes in length of

the filament. The total hysteresis area is a measure of the mechanical work required to convert all bound hRAD51 NPFs between the two ATP-bound states. Using the DNA-binding footprint of a hRAD51 monomer [3 nt (Ristic et al, 2005)] and the estimated protein coverage (80 ± 20)%, we can estimate that the free-energy difference between the two states is 4 ± 1 $k_B$T per hRAD51 protomer (see Appendix for a full derivation). To analyse this hysteretic behaviour under more biologically relevant conditions, experiments were repeated under buffer conditions that allow for ATP hydrolysis within the hRAD51 filaments (Carreira et al, 2009; Figs EV3 and EV4A–C); we observed a similar hysteretic behaviour, which decreased exponentially at the same rate as the rate of hRAD51 dissociation from the ssDNA (Fig EV4B and C). This indicates that the size of the hysteresis area depends on the amount of hRAD51 bound to the ssDNA substrate. We thus conclude that ATP-bound hRAD51-ssDNA filaments can switch between two states that represent a more compact and a more extended conformation.

From analysis of the hysteresis curves, it is apparent that forces of at least 10 pN are needed to induce the conformation switch in the ATP-bound filament (Fig 3D). Above ~60 pN, the extension and relaxation curves overlap, suggesting that all ATP-bound NPFs have completely switched to the extended conformation and transitions no longer occur. When hRAD51-ssDNA was relaxed and subsequently extended again, while keeping the force above 10 pN, the relaxation and extension curves overlapped, showing no hysteresis (Fig 3E), implying that no transitions back to the compact state had occurred. Only at lower forces (~5 pN), the hysteresis disappeared and the filament switched back to the compact state. The observation that the hysteretic behaviour occurs only in a specific range of force values demonstrates that the extended state is unstable and readily switches back to the more compact state when the DNA is relaxed.

From the stretching curves, we can estimate the contour lengths of a hRAD51 protomer in the ATP-compact and ATP-extended filament state (Fig 3F). We assume that initially, all NPFs are in

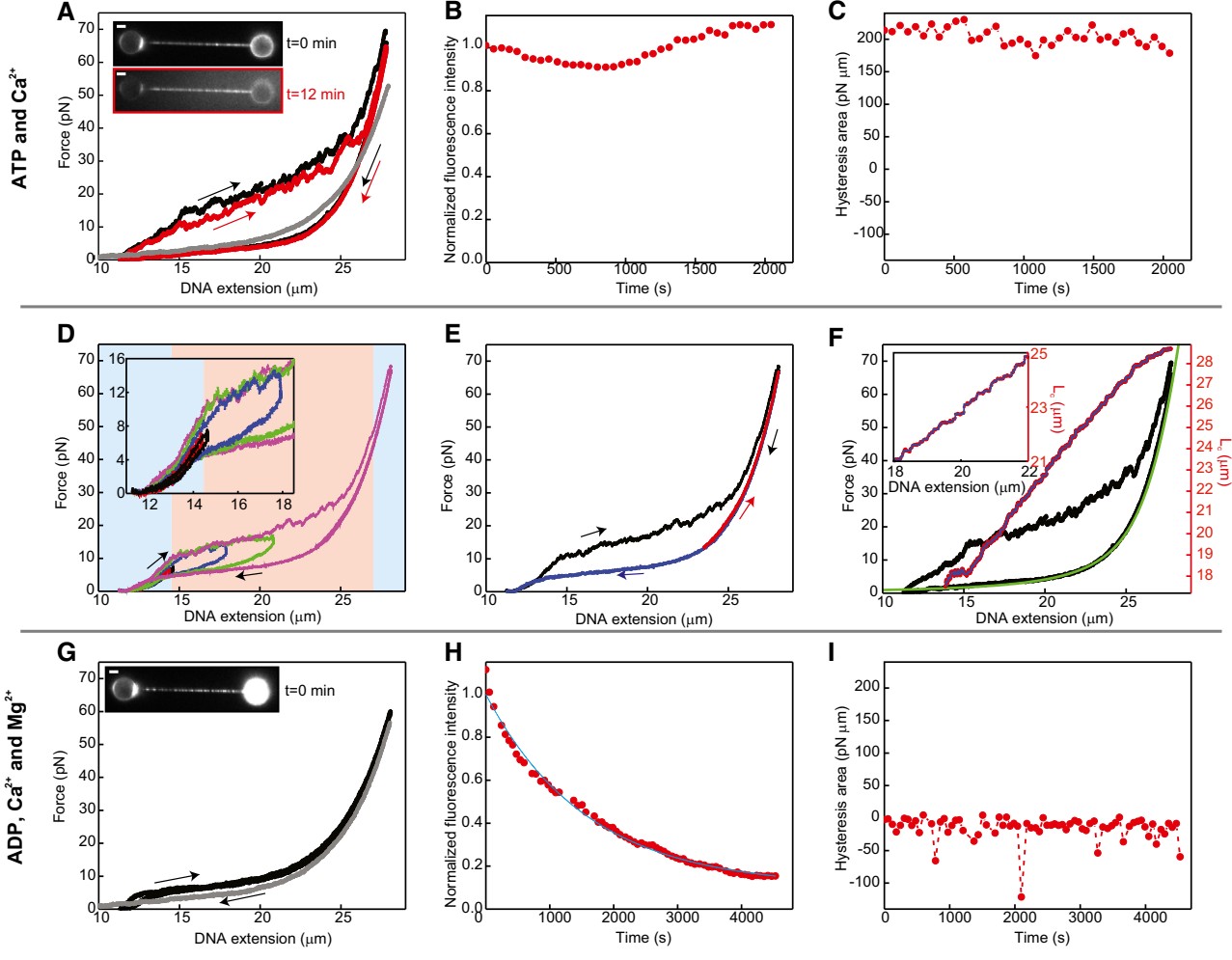

**Figure 3.  Structural transitions within hRAD51-ssDNA NPFs.**

A    Force-extension/force-relaxation cycles of a hRAD51-coated ssDNA molecule in ATP and $Ca^{2+}$ (buffer: 20 mM Tris pH 7.5, 2 mM $CaCl_2$, 2 mM ATP, 1 mM DTT). Time difference between cycles as indicated. Note that extension and relaxation curves of ssDNA (grey dataset) completely overlap. Arrows: direction in which the curves were recorded. Grey: ssDNA. Inset: fluorescence images with edges coloured as force curves. Representative example of two identical experiments. Scale bars: 2 μm.

B    Normalized integrated fluorescence intensity of the same construct over time. The amount of DNA-bound hRAD51 remains constant; that is, photobleaching is negligible during these experiments. Representative example of two identical experiments.

C    The hysteresis area, defined as the area between extension and relaxation curve, remains roughly constant over time under these conditions. Representative example of two identical experiments.

D    Successive extension–relaxation cycles measured in the presence of $Ca^{2+}$, $Mg^{2+}$ and ATP, showing reversible curves up to around 10 pN (first blue region), as all ATP-bound NPFs are in ATP-compact state. At higher forces, ATP-bound NPFs transition to the ATP-extended state and the curves show hysteresis (red region). Above 50 pN, all NPFs are in the ATP-extended conformation (second blue region). Representative example of two identical experiments.

E    The relaxation curve is reversible. After converting all ATP-bound NPFs into the ATP-extended conformation and relaxing the molecule to 24 μm (black), the DNA is extended (red) before complete relaxation (blue). Red and blue curves overlap, showing reversibility of the relaxation curve. Representative example of two identical experiments.

F    Example of an eWLC fit (green) to a relaxation and extension curve (black) and corresponding contour length–extension curve (red) measured in the presence of ATP and $Ca^{2+}$. Representative examples out of 38 experiments. eWLC fit parameters: persistence length 2.501 ± 0.009 nm, stretch modulus 1,599 ± 8 pN and contour length 29.01 ± 0.01 μm (error bars originate from individual eWLC fits). Contour length–extension curves were calculated under the assumption that upon conversion from ATP-compact to ATP-extended conformation, persistence length and stretch modulus remain constant.

G–I    Same as (A–C) but in presence of ADP, $Ca^{2+}$ and $Mg^{2+}$ (buffer: 20 mM Tris pH 7.5, 10 mM Mg(OAc)₂, 2 mM $CaCl_2$, 2 mM ADP, 1 mM DTT). Here, there is no ATP hydrolysis but hRAD51 can disassemble from the ssDNA. We see no hysteresis (G) and (I), while the integrated fluorescence intensity decreases (H) exponentially, yielding a disassembly rate of (4.2 ± 0.3) $10^{-4}$ s$^{-1}$. Representative example of two identical experiments. Scale bars: 2 μm.

Source data are available online for this figure.

ATP-compact conformation; during stretching, these compact filaments undergo conformational changes to the ATP-extended conformation, as can be seen from the saw-tooth-like extension curves. As the relaxation curves are completely smooth, we assume that all NPFs remain in ATP-extended conformation during relaxation and switch back to the ATP-compact state only at very low forces. Thus,

the length of the ATP-extended conformation can be determined from the relaxation curve, by fitting the curves to the extensible worm-like-chain (eWLC) model (Broekmans *et al*, 2016; Fig 3F), which results in an average contour length of the extended ssDNA-hRAD51 filament of $29.01 \pm 0.01$ μm. Using 3 nt as the DNA-binding footprint of a hRAD51 protomer (Ristic *et al*, 2005) and the estimated protein coverage as $(80 \pm 20)\%$, we find that the contribution to the contour length of each hRAD51 protomer in ATP-extended conformation is of $2.2 \pm 0.6$ nm.

In contrast, the contour length of each hRAD51 protomer in the ATP-compact conformation cannot be obtained directly from an extensible worm-like chain (eWLC) fit, since this requires a dataset with all hRAD51 protomers in the ATP-compact conformation. Such a dataset is, however, not available, since the ratio of extended to compact filaments changes along the extension curve and, during relaxation, all filaments are expected to be in ATP-extended conformation. Therefore, a different approach was employed to determine the contour length of the ATP-compact conformation: we assumed that, upon switching from the ATP-compact to the ATP-extended conformation, the persistence length and stretch modulus of the filament remain unchanged and only the contour length changes. In our experimental set-up, the coverage of the ssDNA is only $(80 \pm 20)\%$, and hRAD51 filaments are relatively short. In addition, RAD51 filaments on ssDNA are expected to be very stiff, as is the case for RAD51 filaments on dsDNA (van Mameren *et al*, 2006). Consequently, we expect that the flexibility of the RAD51-ATP-covered ssDNA in these experiments comes mostly from the naked stretches of ssDNA between the small filaments, justifying the assumption that the persistence length and stretch modulus of the complexes do not change upon switching between ATP-compact and ATP-extended conformation. Therefore, the extension curves can be plotted as contour length vs. DNA extension using the persistence length and stretch modulus obtained from the eWLC fits (Fig 3F). This plot shows that the contour length changes in discrete steps (with an average step size of $200 \pm 5$ nm, Appendix Fig S4A and B), corresponding to the conversion of one or multiple NPFs from the ATP-compact to the ATP-extended conformation. Assuming that, at 0 pN, all NPFs are in the ATP-compact conformation, the average contour length of this conformation can be determined from the average contour length before the first step $(17.56 \pm 0.09$ μm). From this, we find that the contribution to the contour length of each hRAD51 protomer in the ATP-compact conformation is $1.4 \pm 0.3$ nm.

Under hydrolysing conditions, ATP is turned into ADP throughout the filament but because hydrolysis is not rate-limiting, filaments normally contain a mixture of ATP and ADP while disassembly is taking place. In order to determine whether the nucleotide cofactor, ATP or ADP, affects the hysteretic behaviour, experiments were repeated in the presence of ADP instead of ATP. Under these conditions, the DNA was densely coated with hRAD51 (Fig 3G, inset) and NPF disassembly occurred at a rate of $(4.2 \pm 0.3) \, 10^{-4} \, s^{-1}$ (Fig 3H), but subsequent extension and relaxation curves overlapped and did not show hysteresis (Fig 3G and I), which suggests that the ADP-bound NPFs adopt a single conformation. However, determining the exact length of filaments in this state was experimentally not possible, since, under these conditions, filament disassembles during the experiment. To estimate the length of the ADP-bound filament, the relaxation curves obtained in the

presence of ATP and ADP were compared (Fig EV5A and B). The curves were very similar, and the small differences observed may be caused by a difference in protein coverage. Therefore, the length of the ADP-bound state is similar to that of the ATP states. Based on this observation, we can also explain why the hysteresis is smaller under conditions of ATP hydrolysis: the parts of the filaments where ATP hydrolysis has already taken place no longer contribute to the hysteresis, thus reducing the fraction of the filament able to switch conformation.

## Structural basis for conformational transitions of the hRAD51-ATP filament

Although much information is now available for the structure of the RecA/RAD51 nucleoprotein filaments (Chen *et al*, 2008; Lee *et al*, 2015; Prentiss *et al*, 2015), our understanding of the mechanistic basis for their unique ability to promote strand exchange between homologous DNA sequences remains incomplete. To provide further insight into hRAD51 function and investigate its apparent ability to adopt different conformations in the polymeric state, we obtained a crystal structure of the hRAD51-ATP filament (Appendix Fig S5A–C and Table S1). Remarkably, the protein crystallized in a monoclinic space group, with two complete turns of helical filament in the asymmetric unit. Thus, unlike previous crystal structures that contained a monomeric or dimeric RecA/RadA/RAD51, related to proximal protomers by hexagonal or trigonal symmetry, our crystal structure yielded information on the conformation of 14 independent hRAD51 molecules, in a filament state and bound to ATP.

In the crystal, ATP-bound hRAD51 forms a continuous right-handed filament with seven subunits per helical turn (Fig 4A and B). As hRAD51 exists in a range of oligomeric species in solution that depend on concentration for their relative abundance, it is likely that crystal growth was seeded by heptameric hRAD51 present in the crystallization buffer. The arrangement of the hRAD51 protomers within the filament and their mode of self-association is similar to what was observed for the filament structure of yeast Rad51 (Conway *et al*, 2004). The heptameric repeat of the helical hRAD51-ATP filament has a pitch of 128.0 Å and a protomer rise of 18.3 Å. These values are within the distribution measured by cryoEM for *bona fide* presynaptic hRAD51 filament structures (Short *et al*, 2016).

Previous crystallographic analysis of the yeast RAD51 filament had revealed the presence of two slightly different interfaces in the asymmetric unit, implying that the functional unit of the filament might be a dimer (Conway *et al*, 2004). Inspection of the 12 independent dimer interfaces in the two heptameric turns of the hRAD51-ATP filament structure showed the presence of two distinct dimer conformations, alternating along the filament (Fig 5A), in a qualitatively similar arrangement to what had been observed for yeast Rad51 (Conway *et al*, 2004). Comparison of the two dimer types by superposition shows that, relative to the first hRAD51 protomer in the dimer, the second protomer undergoes a rigid-body movement comprising of a slight tilt towards the filament axis coupled to a small increase in twist (Fig 5B). The structure of the individual RAD51 subunits remains unchanged in the two dimer forms of the filament. The pivot point for this composite rotation is centred at the dimer interface,

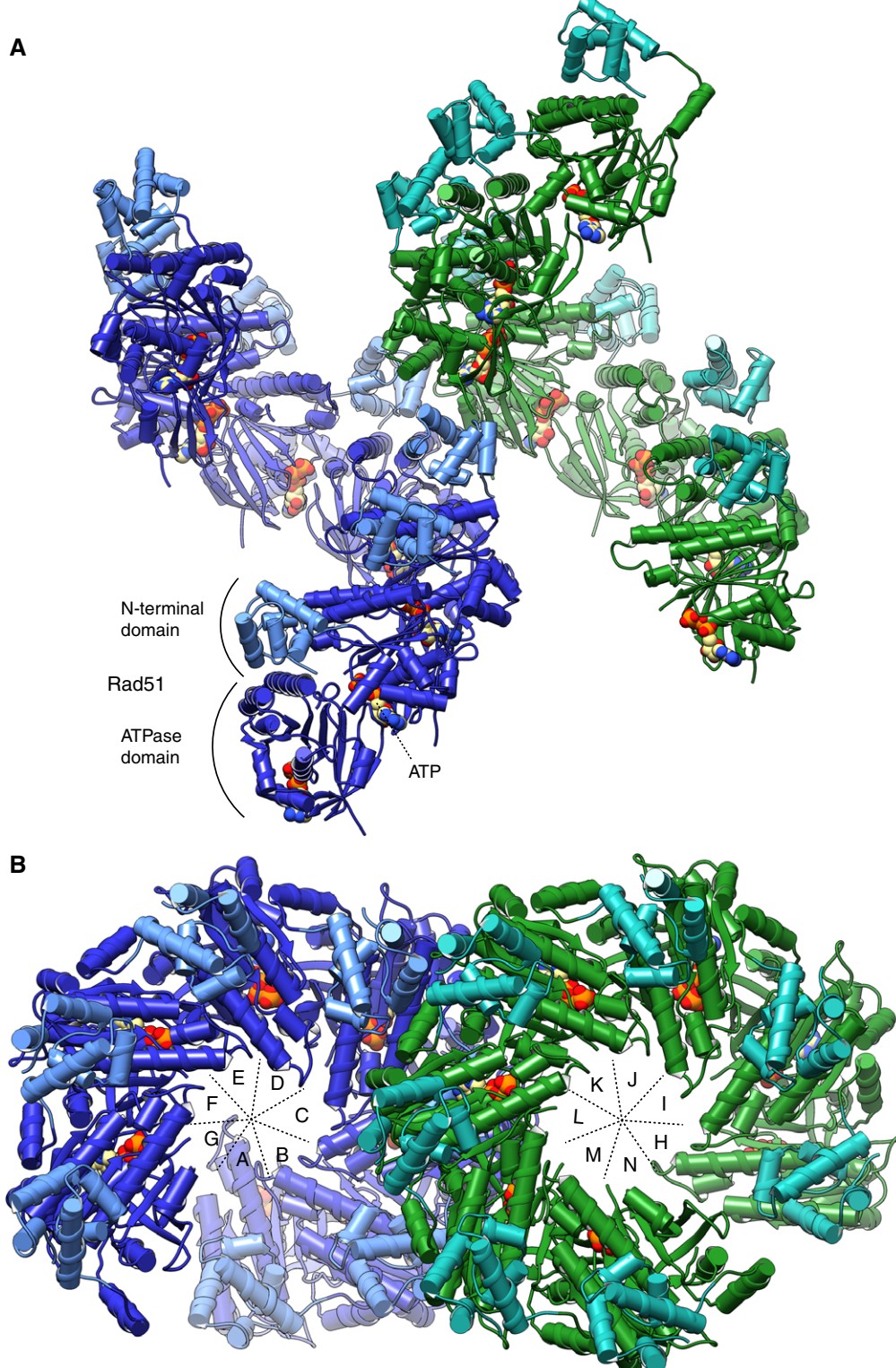

**Figure 4. Crystal structure of the human hRAD51-ATP filament.**

A, B   Side (A) and top (B) views of the asymmetric unit content of the crystal, which consists of 14 ATP-bound copies of hRAD51 arranged as two heptameric right-handed filaments. The two heptameric oligomers form continuous helical filaments running through the crystal. The hRAD51 molecules are drawn as ribbons, colour-coded in blue or green in the two heptamers, with cylinders marking the position of each alpha helix. The ATP molecule is drawn as spacefill model. The positions of the N-terminal and ATPase domains of RAD51 are indicated in (A). The RAD51 chains in the two heptameric assemblies are labelled A to G and H to N in (B).

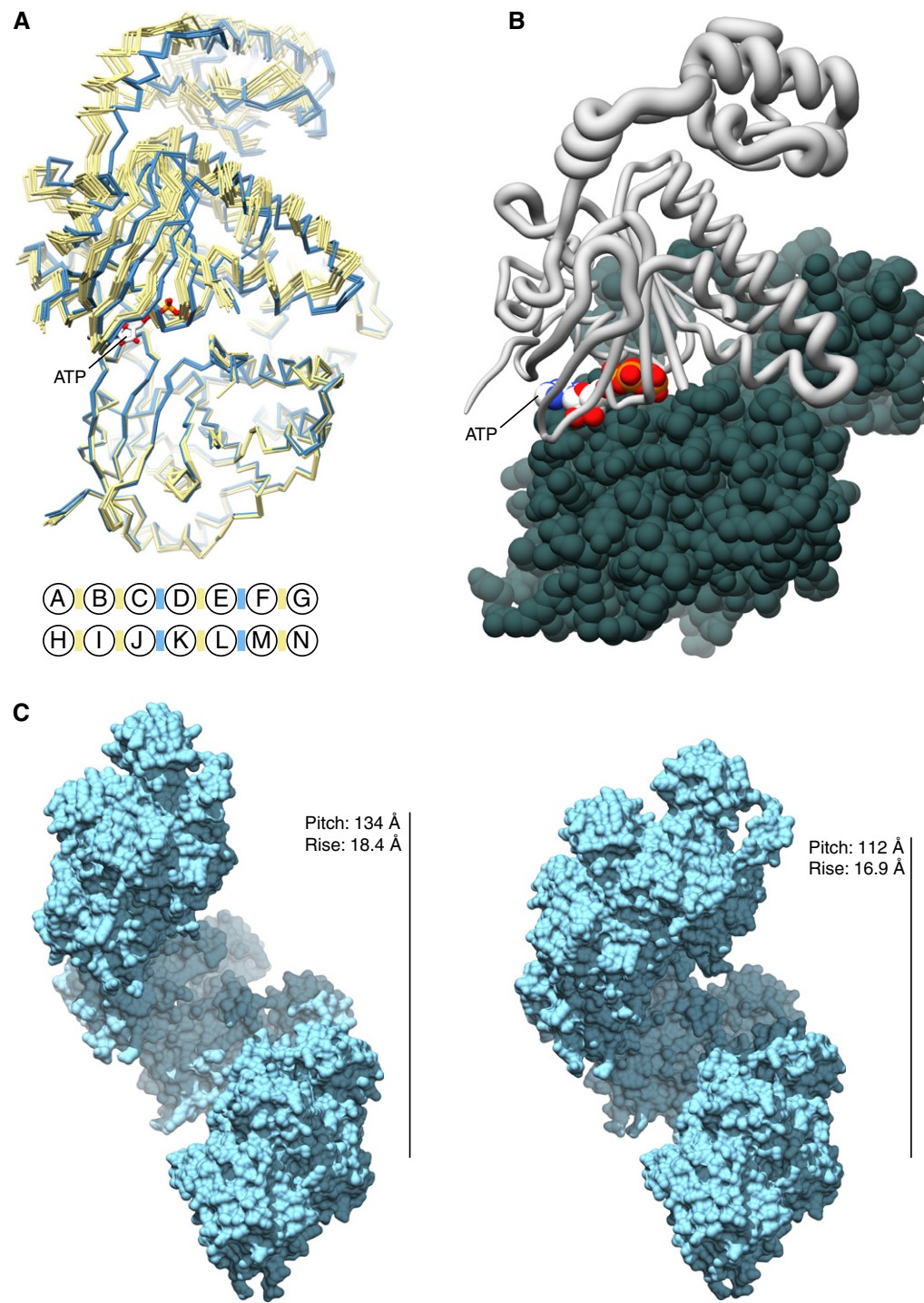

**Figure 5.  Two dimer interfaces exist in the crystal structure of the hRAD51-ATP filament.**

A   Structural superposition of the 12 dimers present in the asymmetric unit of the hRAD51-ATP filament. Each hRAD51 dimer is coloured in light blue or yellow,
    according to the interface type between its two protomers. The protein chains of the dimers are drawn as a $C_\alpha$ backbone, with one ATP moiety shown as stick model.
    The illustration underneath the superposition reports the distribution of interface types between protomers in the two hRAD51-ATP heptamers, with RAD51 chains
    labelled A to G and H to N, as in Fig 4B. Each interface is marked by a coloured bar, colour-coded blue or yellow as in the structural superimposition.
B   The relative displacement in the position of hRAD51 residues between the two dimer types is illustrated by drawing the hRAD51 $C_\alpha$ backbone as tube of varying
    radius, in direct proportion to the rmsd value for each amino acid (a larger radius corresponds to a higher rmsd). The structure of the reference hRAD51 structure
    used in the superposition is drawn as a spacefill model.
C   Two filament structures of different pitch and rise are obtained by modelling a filament based exclusively on one or the other dimer type found in the crystal
    structure of the hRAD51-ATP filament (see also Movies EV1 and EV2). Seven hRAD51 protomers corresponding approximately to one helical turn are shown in both
    cases, drawn as molecular surfaces in light blue.

with residues surrounding the ATP-binding site displaying the least movement.

The occurrence of two distinct dimer conformations across 12 independent filament interfaces, rather than a single interface type or a range of different interfaces, is striking and suggests that they might be functionally relevant. Modelling of hRAD51-ATP filaments constituted entirely of one or the other dimer type yields filaments with different pitch and rise (134 and 18.4 Å vs. 112 and 16.9 Å), reflecting a looser (7.3 hRAD51 chains/turn) or tighter (6.6 hRAD51 chains/turn) winding of the helical hRAD51 assembly around the filament axis (Fig 5C). The values for the rise of the two filament models, at 1.84 and 1.69 nm, fit within the contour lengths measured for the two states of the hRAD51-ATP filament in the optical tweezers (2.2 ± 0.6 and 1.4 ± 0.3 nm). The narrower difference in rise for the two RAD51 dimer types in the crystal of the ATP-bound filament, relative to the rises measured in solution, might be due to the constraint of the crystal lattice, as well as to the absence of ssDNA. Interestingly, morphing between the two filament models mimics a structural transition reminiscent of a peristaltic movement, whereby the filament cycles between a more compact, overwound state and a more loosely wound, extended state (Movies EV1 and EV2).

## Discussion

The knowledge that the hRAD51 NPF possesses remarkable conformational plasticity has been available for a long time, but the correct mechanistic interpretation of this filament property still eludes us. In this study, converging evidence from single-molecule and structural experiments shows that ATP-bound hRAD51 NPF exists in two distinct forms that differ in pitch and protomer interface. Moreover, the data indicate that the simultaneous presence of two NPF conformations requires ATP and cannot be realized with ADP.

Our single-molecule analysis shows that hRAD51-ssDNA filaments can exist in different nucleotide-dependent states (Fig 6): (i) an ATP-extended state, (ii) an ATP-compact state, and (iii) an ADP-bound state that precedes disassembly. What is the mechanism for the conversion between different filament states?

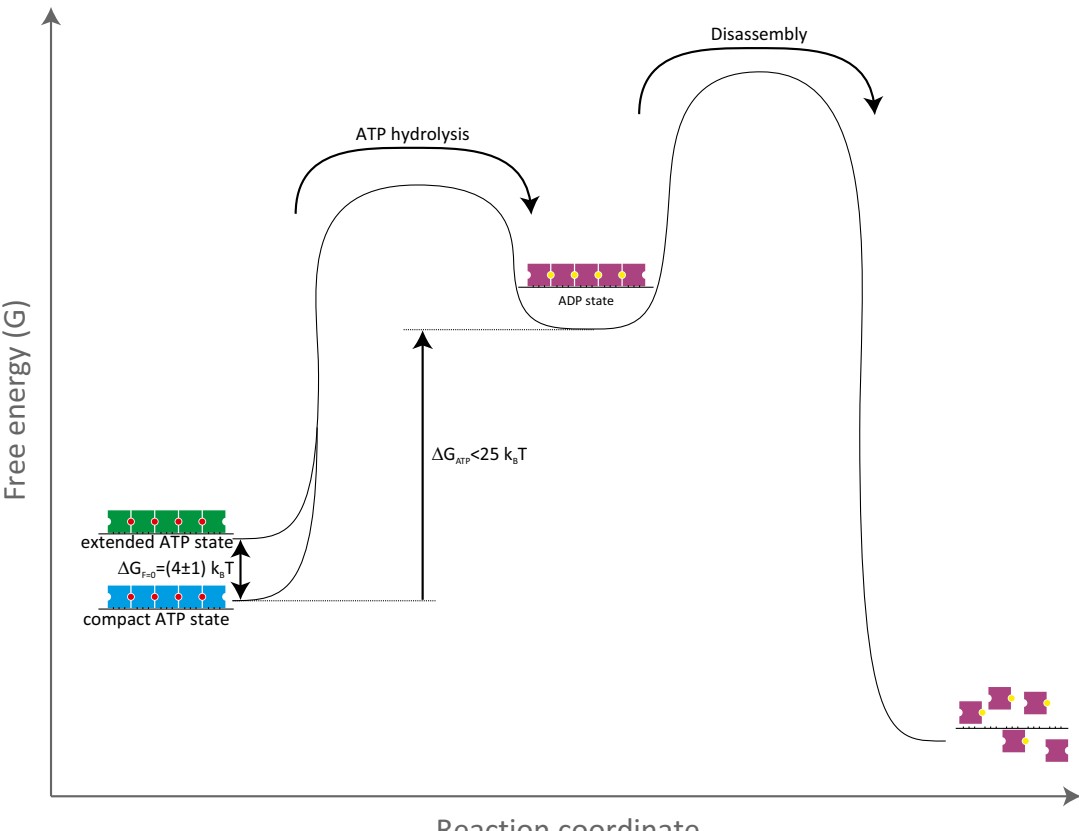

**Figure 6. Cartoon model of different states of hRAD51 on ssDNA.**

Based on the experiments involving stretching cycles such as shown in Figs 3A–I, EV3, and 5G and H, we propose a cartoon model with the following conformational states and possible transitions. Because of the large hysteresis in the data shown in Fig 3A–C, we propose that there are two ATP-bound states: an ATP-compact and an ATP-extended state with a free-energy difference of $(4 \pm 1)$ $k_BT$. Because there is no significant hysteresis in the experiments shown in Fig 3G–I, we propose that there is only one ADP-bound state, from which the hRAD51 NPF can disassemble: the ADP state. For switching between ATP-bound and ADP-bound states, additional energy is required. This is generated by the hydrolysis of one ATP molecule (providing ~25 $k_BT$). Black: ssDNA, blue: ATP-compact hRAD51 monomer, green: ATP-extended hRAD51 monomer, purple: hRAD51 monomer in ADP state, red: ATP, yellow: ADP.

                                                                                       

Switching between the ATP-compact state (with a contour length of $1.4 \pm 0.3$ nm per hRAD51 monomer) and the ATP-extended state (with a contour length of $2.2 \pm 0.6$ nm per hRAD51 monomer) can be triggered by force but given the small energy difference between the two ($4 \pm 1$ $k_BT$ per hRAD51 monomer), the extended state should be accessible simply by thermal excitation. Conversely, under experimental conditions where ATP hydrolysis can occur, the filaments can switch to a single disassembly-competent ADP state from both the ATP-compact and the ATP-extended state, since the total disassembly rate does not depend on ssDNA template tension. Disassembly can only occur from the ADP-bound state, and not from either of the ATP-bound states, because, just as was shown previously for dsDNA (van Mameren *et al*, 2009b), disassembly of RAD51 from ssDNA does not happen in conditions that do not allow ATP hydrolysis (i.e. in a buffer containing no $Mg^{2+}$). ATP hydrolysis must thus precede disassembly, but, as the total disassembly rate does not depend on the nucleotide cofactor initially bound within the filament, it is not rate-limiting. The RAD51 disassembly rates measured under different experimental conditions are given in Appendix Table S2. The free-energy difference between a hRAD51 monomer in either of the ATP states and a hRAD51 monomer in the ADP state is at most 25 $k_BT$, as this is the energy provided by ATP hydrolysis per ATP hydrolysed (Lodish *et al*, 2004). This conversion might, as was previously suggested for RecA-ssDNA NPFs (Kim *et al*, 2014), involve a cooperative transition between neighbouring monomers.

On the basis of electron microscopy experiments, it has been proposed that hRAD51-ssDNA NPFs are characterized by a high degree of conformational freedom (Yu *et al*, 2001). This is in line with our results from both single-molecule and crystallography analyses, showing the occurrence of different hRAD51-ssDNA filament states. Other previous work on hRAD51-dsDNA (Atwell *et al*, 2012) and RecA-ssDNA (Kim *et al*, 2014) has also shown similar transitions within NPFs. Those studies, however, suggested the existence of only two states: an ATP-bound extended state and an ADP-bound compressed state. Our data clearly indicate that, next to the ADP-bound state, the NPF can exist in two ATP-bound states that differ in contour length. Remarkably, the reported hRAD51-dsDNA filament length of 1.5 nm per hRAD51 monomer (Ristic *et al*, 2005) matches that of the ATP-compact state for hRAD51-ssDNA ($1.4 \pm 0.3$ nm); this correspondence might be significant for homology search and strand exchange processes during homologous recombination. Interestingly, the work on RecA-ssDNA (Kim *et al*, 2014) reported significant reloading of ATP within the filament. However, when we tested whether reloading of ATP contributes to the hysteretic behaviour that we observe, by allowing ATP to be present during NPF formation but not during the extension–relaxation cycles (Fig EV4D–F), we found no significant effect of ATP reloading.

In this study, we have provided important new insights into the dynamics of hRAD51-ssDNA filaments, with particular emphasis on the role played by the nucleotide cofactor ATP. Our combined evidence from single-molecule and structural experiments reveals that the ATP-bound hRAD51 NPF is a highly flexible entity. Building on existing evidence for variations in filament pitch, we now demonstrate that the RAD51-ATP-ssDNA filament can exist in two specific, interconvertible conformational states.

The ensuing structural plasticity provides an appealing molecular basis for the reactions of homology recognition and strand exchange that underlie homologous recombination. Thus, we propose that the concerted conversion between the two filament states described here, when propagating through a synaptic filament, might represent a critical aspect of the search for homologous DNA sequence. The small amount of energy needed for interconversion between ATP-bound states, relative to that required to reach the ADP-bound state ($4$ $k_BT$ vs. $25$ $k_BT$), would facilitate disengagement during incorrect pairing events, as the filament cycles between the ATP-bound states in the search for homology. Finally, we note that specific conformational transitions, such as the ones we describe, might be promoted by recombination modulators inside the cell. Indeed, recent work on RAD-51, the worm orthologue of human RAD51, has indicated that the RFS-1/RIP-1 complex acts by remodelling the RAD-51 presynaptic filament into a more open, flexible conformation that stimulates strand exchange (Taylor *et al*, 2015, 2016).

# Materials and Methods

### DNA constructs for trapping experiments

The preparation of the construct which can be used for ssDNA experiments upon force-induced melting was described previously (Candelli *et al*, 2013). In brief, biotinylation of both the 3′ and 5′ end of the same DNA strand is achieved by sequential annealing and ligation of oligonucleotides (5′-ggg cgg cga cct gga caa-3′ and 5′-agg tcg ccg ccc ttt ttt tTt TtT-3′) to first biotinylate the 5′ end and subsequently the annealing and ligation of an oligonucleotide (5′-TtT tTt ttt ttt aga gta ctg tac gat cta gca tca atc ttg tcc-3′) to the 3′ end of a linearized Lambda DNA (48,517 nt) molecule (T = biotinylated).

### Experimental conditions

Catching of the beads (4.5-µm streptavidin-coated polystyrene microspheres) and the DNA were performed in PBS buffer, consisting of 10 mM phosphate and 150 mM sodium chloride at pH 7.3–7.5. DNA melting for generation of ssDNA templates was performed in 20 mM Tris, pH 7.6. Buffer conditions in the protein incubation and imaging channels were indicated in the text.

### Set-up combining optical trapping, fluorescence microscopy and microfluidics

The custom-built experimental set-up was described in detail elsewhere (Gross *et al*, 2010). Briefly, it is built around a Nikon inverted microscope equipped with a 1,064-nm trapping laser, where the two traps that can be manipulated independently using steerable mirrors are generated by splitting the laser into perpendicularly polarized beams using a half-wave plate and polarizing beam splitter. Using a second polarizing beam splitter, the two trapping beams are recombined and coupled into a water-immersion objective on the microscope. By collecting the transmitted light using an oil-immersion condenser and rejection of the unwanted light by a third polarizing beam splitter, the force can be detected on a position-sensitive

diode. The bead-to-bead distance was measured using real-time template matching of bright-field images obtained by blue LED illumination. For fluorescence imaging of hRAD51-Alexa 555, a 532-nm excitation laser was simultaneously coupled into the microscope and imaged on an EMCCD camera. To enable fast buffer exchange between beads, DNA, buffer and protein channels, a custom-made (u-Flux, LUMICKS B.V.) multichannel laminar flow cell was mounted on the microscope stage.

**Derivation of free-energy difference from force-extension curves**

The free-energy difference per hRAD51 monomer between the compact and extended ATP states is calculated from the fluorescence intensity. As a starting point, the free energy per hRAD51 monomer is written as follows:

$$\Delta G = \frac{A}{k}.$$

Here, $\Delta G$ is the free-energy difference between the two states, $A$ is the measured hysteresis area, and $k$ is the number of bound hRAD51 monomers. The value of $k$ is calculated from the fractional coverage of the DNA template ($v$) and the number of available binding sites ($n$) as follows:

$$k = vn.$$

Furthermore, $n$ can be determined from the binding site size or footprint of hRAD51 on ssDNA ($m$) and the number of bases in the DNA template used ($N$), as follows:

$$n = \frac{N}{m}.$$

Combining these three formulas yields:

$$\Delta G = \frac{Am}{vN}.$$

To calculate the free-energy difference between the ATP-extended and compact states, the following values are used:

(1)    The hysteresis area $A$ is determined from Fig 3C as $205 \pm 2$ pN μM.
(2)    The binding site size $m$ is three bases (Ristic *et al*, 2005).
(3)    The fractional coverage $v$ in our experiments is estimated to be $(80 \pm 20)\%$.
(4)    The number of bases in the DNA template ($N$) is 48,517.

These numbers yield an estimated free-energy difference of $(4 \pm 1)$ k$_B$T per hRAD51 monomer.

**Methods for protein purification and labelling**

Human full-length hRAD51 was expressed in *Escherichia coli* and purified using a new protocol that exploits the affinity of hRAD51 for BRCA2 BRC-repeat 4 (https://www.addgene.org/105045/). Briefly, full-length, human RAD51 was co-expressed with a BRCA2 BRC4 sequence fused to an N-terminal dual His-MBP tag in the BL21(DE3) Rosetta2 *E. coli* strain. After initial purification of the His-MBP-BRC4–RAD51 complex by Ni$^{2+}$-NTA chromatography, RAD51 is separated by the His-MBP-BRC4 fusion protein using heparin–Sepharose chromatography, as RAD51 binds to the column in low-salt conditions, whereas the His-MBP-BRC4 protein remains in the flow through. RAD51 is further purified by gel-filtration chromatography, concentrated and stored in aliquots at −80°C. This method allows for the rapid and efficient recovery of milligram amounts of purified RAD51 from 1 l of BL21(DE3)Rosetta2 cells.

hRAD51 (isoform K313, variant C319S) fluorescent labelling with Alexa Fluor 555 was performed as was described previously (Modesti *et al*, 2007). The degree of labelling was estimated to be around 80%. Biochemical characterization showed that hRAD51 (C319S) is proficient in ATP hydrolysis, strand exchange and DNA binding (Appendix Fig S1A–C).

**Crystallization and X-ray crystal structure determination**

hRAD51 was crystallized using the hanging-drop diffusion method. 2 mM MgATP was added to the protein samples shortly before crystallization. Drops were set up in a 1:1 ratio of protein (7.4 mg/ml) and mother liquor, which contained 0.1 MES pH 5.2 and 22% v/v MPD, at 293 K. Initial crystals were small and were therefore improved with streak seeding, where 9% v/v sucrose was added to the mother liquor. X-ray diffraction data were collected at the PROXIMA1 beamline of the SOLEIL synchrotron, Gif-sur-Yvette, France. The data were processed in XDS and Aimless, and the structure was solved by molecular replacement (MR) in Phaser, using the coordinates from PDB entry 1N0W as search model. hRAD51-ATP crystallized as a helical filament in space group P2$_1$, with unit cell: 117.7 Å, 128.0 Å, 230.1 Å, 90, 90.3, 90 and two heptameric assemblies in the asymmetric unit. For structure determination by MR, analysis of crystal cell content showed the presence of several RAD51 molecules in the asymmetric unit, so initial searches in Phaser were performed looking for multiple copies of the RAD51 ATPase domain (PDB id 1N0W). MR solutions were scored as successful when two RAD51 ATPase domains were juxtaposed in a manner that was in agreement with the known association mode of RAD51 protomers in the structure of the yeast RAD51 filament (PDB id 1SZP). The successful solutions were retained as fixed solutions in successive rounds of MR in Phaser. By iteration of the process, we were able to obtain the position of all 14 chains in the asymmetric unit. A full-length model for human RAD51 was built, using PDB entry 1SZP for the RAD51 N-terminal domain and 1N0W for modelling hRAD51's interdomain linker sequence. The structure was refined in Phenix (Adams *et al*, 2002) to a resolution of 3.9 Å, using Coot (Emsley *et al*, 2010) for model building and applying NCS restraints to the ATPase and N-terminal domains of the 14 hRAD51 molecules in the asymmetric unit.

**Data availability**

The structural coordinates of the refined model and the structure factors have been deposited in the RCSB Protein Data Bank (http://www.rcsb.org/) and assigned the identifier 5NWL.

**Expanded View** for this article is available online.

## Acknowledgements

We would like to thank Michael A. Longo for developing the hRAD51 purification protocol, based on RAD51's affinity for BRCA2 BRC4. We would like to thank Joseph Maman for helpful discussions and critical reading of the

manuscript. This work was supported by VICI grants (G.J.L.W. and E.J.G.P.) from the Nederlandse Organisatie voor Wetenschappelijk Onderzoek, an European Research Council starting grant (G.J.L.W.), the French National Cancer Institute (Grant PLBIO13-103) (M.M.), the ARC Foundation for Cancer Research (M.M.), the A*MIDEX Project (no. ANR-11-IDEX-0001-02) and the "Investissements d'Avenir" French Government programme (M.M.); a postdoctoral fellowship of the Fondazione Cenci-Bolognetti to T.M.; a Wellcome Trust investigator award (104641/Z/14/Z) to L.P.; and a postdoctoral fellowship of the Fondation Aix-Marseille Université to E.B.G.

## Author contributions

IB and AC performed the single-molecule experiments. TM and LP performed the crystallization and X-ray crystal structure determination. EBG, MM and TM purified the proteins and performed the biochemical analysis of the protein samples. IB, LP, GJLW and EJGP wrote the manuscript. MM, LP, GJLW and EJGP conceived the project and led the research. All authors discussed the results and commented on the manuscript.

## Conflict of interest

The authors declare that they have no conflict of interest.

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
