## [Review Process File · The EMBO Journal]

Two distinct conformational states define the interaction of human RAD51-ATP with single-stranded DNA

Ineke Brouwer, Tommaso Moschetti, Andrea Candelli, Edwige B. Garcin, Mauro Modesti, Luca Pellegrini, Gijs J.L. Wuite & Erwin J.G. Peterman

Review timeline:

Submission date:	10 September 2017
Editorial Decision:	14 November 2017
Revision received:	1 January 2018
Editorial Decision:	15 January 2018
Revision received:	7 February 2018
Accepted:	8 February 2018

Editor: Hartmut Vodermaier

Transaction Report:

Pre-decision Consultation - Editor

13 October 2017

Thank you for submitting your manuscript on RAD51 conformations on ssDNA to The EMBO Journal. We have now received the reports from three experts, which I am enclosing copied below. As you will see, all reviewers appreciate your approaches and consider various parts of your results potentially very interesting. However, they also raise several major concerns regarding presentation, experimental description and interpretation of the data, but importantly also regarding the rather loose connections between the different lines of investigation in the paper, and the unclear biological significance of the present findings.

Since especially the last two concerns would in our view have a major impact on the suitability of this study for a broad general journal such as The EMBO Journal, it would be important to hear whether there would be any concrete strategies by which you could address these key concerns if given the opportunity to revise the manuscript. I would therefore like to invite you to discuss the attached reports with your coworkers, and to then draft a tentative point-by-point response detailing how you could envision addressing the referees's comments. As already said, especially strengthening the evidence for biological significance and better connecting the different types of newly obtained data would be important factors in our final decision on whether further consideration for The EMBO Journal would be warranted in this case.

I would appreciate if you could send us such a tentative response letter (parts of which we may choose to share and discuss with the referees) during the course of the coming week. Should you have any further questions in this regard, of course please do not hesitate to let me know.

REFeree REPORTS

Referee #1

(Report for Author)

The authors show that ATP-bound human (hRAD51) can exist in two different conformational states via a combination of single-molecule optical trapping and x-ray crystallography. This study is technically well executed and provides an intriguing insight into RAD51-ssDNA filament dynamics, albeit with limited discussion on the implications for *in vivo* recombination. Another strength of this study is that the single-molecule biophysics is married with a new crystal structure of the hRAD51 filament (in the absence of DNA). While overall sound, the analysis had relatively low statistics (~4 molecules) for some of the conclusions. I could also not follow a few technical details and several figures were mislabeled. In sum, I think this manuscript reports a significant finding that will be broadly interesting to readers of this journal. Below, I include suggestions that will improve clarity and possibly increase the biological significance:

Main points:

- Is RAD51 disassembly sequence dependent? Some puncta seem to dissociate much faster than others in Fig 1 & 2. The high AT-content in half of their DNA substrate could let the authors analyze whether disassembly is faster/slower there.
- Does figure 1E/2B/3A-C show fits to a single ssDNA-RAD51 filament? An average of at least a half dozen molecules with appropriate estimates of the uncertainty from multiple replicates (S.D. or 95% C.I.) will be more useful.
- I couldn't follow the logic for the 4 kBT measurement reported in relation to Figure 3. Please clarify how this number was derived. This is an important point that should be included in the supplement.
- The authors assume that the persistence length and stretch modulus of the filament do not change between the two states. This assumption should be rationalized more carefully.
- Include a table of relevant crystallographic information related to Figure 4.
- In supplementary figure 3D, the data shown with error bars seem that there is no significant difference between ADP and ATP on the disassembly. Clearly mention how many molecules were measured, how statistical test was performed to conclude their argument.
- The difference between hRAD51-ssDNA filaments and bare ssDNA was difficult to see in figure 3A. Please clarify.
- Figure S3 shows that ADP-bound RAD51 is more stable than ATP-bound RAD51. This is in direct disagreement with the reference cited (Ref. 19) that the ADP-bound is an intermediate to disassembly. This needs to be clarified.
- The goodness of fit on figure S3C needs to be clarified.
- I believe that the captions in Figs S5 and S6 actually relate to Fig 3, not Fig 4 (as written).
- The authors should speculate more about the biological significance of the two states. Do these possibly participate in heteroduplex rejection? BRC-repeat interactions?
- The authors use a new RAD51 purification protocol. This should be explained in the methods

Minor points:

- In Figures 1 - 3, quantification of more molecules with appropriate statistical tests with reinforce conclusions.
- Multiple figure panels frequently change axis dimensions although displaying similar data (Ex. Figure 3E/F).
- Add letters used in Figure 5A to Figure 4 to show the different RAD51 subunits in the crystal structure.
- The text mentions that "there was significant hysteresis between the curves", but this sounds strange because the two curves together means hysteresis. It would be more appropriate to say "the force-extension curve shows hysteresis".

Referee #2

(Report for Author)
 Review on Brouwer et al.
 EMBO-2017-98162

The paper by Brouwer et al. describes single-molecule analysis of human RAD51 nucleoprotein filaments and crystal structure analysis of human RAD51 filament. RAD51, a homolog of bacterial RecA, is essential for homologous recombination in eukaryotes. RAD51 is bound to single-stranded (ss) DNAs to form nucleoprotein filament (NPF). In the presence of ATP, RAD51 NPF is active for homology search and strand exchange with homologous double-stranded (ds) DNAs. To know the molecular mechanism of homology search in homologous recombination, it is important to decipher the dynamics of RAD51-NPF bound to ATP. The paper consists of three inter-related studies of RAD51-NPF. First, the authors analyzed dissociation dynamics of single RAD51 NPF and found that dissociation of RAD51 from ssDNAs is independent of tension of ssDNA. This is different from those of RAD51 from dsDNAs, which is dependent of the tension (as shown by one of the authors' group previously). Second, by applying forces for extension and contraction on the RAD51-NPF, the authors revealed a novel physical property of ATP-bound RAD51-NPF (not ADP-bound form), which shows "hysteresis", suggesting the presence of two different physical states of RAD51 protomers in the NPF, which are likely to be extended and contracted forms. The transition might be unidirectional, once becomes the extended form, the extended form does not become a contracted form. Third, they also determined a crystal structure of human RAD51-ATP filaments (without any DNAs in it) and found that two interfaces of RAD51 protomers. This is consistent with a previous report of yeast Rad51 filament. These results are very interesting. Particularly hysteresis in ATP-bound RAD51 NPF is very much novel physical property which would be seen in other protein machinery. The results in the paper might be suitable for publication in EMBO journal. However, there are some concerns, which should be addressed prior to publication.

One major concern is that interpretation of the connection of two results; hysteresis properties obtained by single-molecule analysis and two interfaces of RAD51 monomers. To support this, the authors need more analysis. For me, most simple (maybe difficult) one is to determine the crystal structure of RAD51 filament in the presence of "ADP". Since ADP-bound RAD51 filament does not show hysteresis (Fig. 3G), the authors would expect single interface in ADP-bound RAD51 filament. If this kind of result is provided, although this is still just additional correlation, the authors may strengthen the main conclusion.

Minor points:

1. Page number should be provided.
2. Figure 1: The authors should analyze the variations of dissociation of RAD51 at a local level in a way to address whether cooperative or processive dissociation of RAD51 filaments from the end occurs or not.
3. To make general readers understand hysteresis, it would be better to write the property of ssDNA alone (in a separate figure from Fig. 3A), which shows no hysteresis. Then, the authors should compare ATP-bound and ADP-bound RAD51-ssDNA in terms of hysteresis (Fig. 3A versus 3G) prior to indicating the hysteresis properties of ATP-bound RAD51-ssDNAs.
4. For structure determination of RAD51-ATP filaments, how the authors add ATP in the crystal. In method, at least in hanging-drop buffer, no description of ATP. Do they soak RAD51-crystal with ATP?
5. Figure 5A and legend: There is clear gap between the figure and the legend. Backbone ribbons are pale blue and red, but in the legend, green and red. I can not understand what A-H means in the bottom of Figures. The legend said different chains, but no 7 blue chains init.

Referee #3

(Report for Author)
 This manuscript combines two different biophysical characterizations of human Rad51: single-

molecule force-extension studies of Rad51 - ssDNA filaments, and a crystal structure in the presence of ATP.

While the crystal structure is a truly impressive feat, the overall message of this work is confusing. The force-extension curves are interpreted to reflect two differently-extended filaments for the hRad51-complex, and described as transitioning cooperatively between them. The crystal structure does show two slightly different protein-protein interfaces in the filaments, alternating. That they alternate is very intriguingly similar to the yeast Rad51 crystal structure, but it doesn't seem to connect well with the in vitro data.

The interpretation of the single-molecule data is confusing. Could the hysteresis simply reflect a tendency of short hRad51-ATP filament segments to interact with one another (or with free ssDNA segments, in a pseudo strand-pairing reaction)? Why is the shorter filament derived modelled from the crystal structure assigned to one of the single-molecule-determined types when its length is really within error of both (1.69 μ M per protomer from the crystal vs. 1.4 \pm 0.03 and 2.2 \pm 0.6)? Why is it reasonable to assume, when determining these lengths from the single molecule data, that the persistence length of the two types would be the same?

Why is figure S3 titled "disassembly ... depends on the presence of ADP or ATP ..." when no nucleotide-free data is shown, and the two rates determined in this figure seem to be the same within error (and are referred to as the same at some point in the main text)?

Given that disassembly rates for the ADP- and ATP-bound states appear to be similar, why does figure 6 and its discussion assume that ATP hydrolysis must precede disassembly?

What is the biological significance of a pathway for hRad51-ssDNA filament disassembly?

Figure 6 includes a confusing reference to BRC4 that needs a bit more context.

It would help the reader to include a summary chart listing all the different rates and lengths determined in the various assays.

Finally, the crystal structure, while it appears to be carefully refined, needs a bit more documentation:

- 1) The protein purification method is described simply as "unpublished". Given recent pushes for reproducibility in publications, the full methodology for reproducing this protein sample should be described.
- 2) No mention of ATP is given in the methods - it just shows up in the picture. Please describe how much ATP was added, whether Mg⁺⁺ or Ca⁺⁺ were present, and be explicit about how it was kept from hydrolyzing during crystallization.
- 3) The molecular replacement problem (orienting 14 protomers with a partial search model at \sim 4Å resolution) sounds truly daunting, and it would be nice for others with tough problems if a more were said about how it was tackled.
- 4) Only a 2Fo-Fc (weighted?) is shown. Due to the strong possibilities of model bias at such low resolution and with no experimental phases, it would be nice to see some minimally-biased omit maps.
- 5) Are the details of the two interfaces seen in this work similar to those seen for yeast Rad51, or just the concept of two alternating interfaces?
- 6) (just a typo) in the methods section "... model for human ATP" should be "... the human Rad51-ATP complex"

Pre-decision Consultation - Authors

25 October 2017

Thank you for your message of October 13th, 2017 about our manuscript "Two distinct conformational states define the interaction of human RAD51-ATP with single-stranded DNA". We are pleased that the reviewers agree that our findings are interesting and suitable for publication in The EMBO Journal. Here we would like to take the opportunity to address the comments of the reviewers as well as address the points you raised about our manuscript.

We have attached a preliminary point-by-point reply to the reviewers' comments. As you will see, many of the points raised can be addressed straightforwardly. As you indicated, two important questions deserve special attention: the biological significance of our findings and the connection of the two different types of data that we present in the manuscript.

With respect to the connection of the single-molecule data to the structural data, we propose to explain in more detail the physical implications of the hysteresis that we observed in our single-molecule experiments and how they relate to our structural observations. Our argument is as follows:

- Our single-molecule force-extension data shows clear hysteresis between extension and relaxation curves. This hysteresis is best understood in terms of force-induced conformational changes of the RAD51-ssDNA nucleoprotein filament that result in reversible changes in the length of the filament. From our force-extension curves we can estimate the length difference of the two conformations and the mechanical work required to go from compact to extended conformations.
- Our crystal structure of the RAD51-ATP filament shows the presence of two different and specific protein-protein interfaces between adjacent RAD51 molecules, which is strikingly reminiscent of what previously observed for the filament structure of yeast Rad51 (Conway et al, NSMB, 2004). When we build two models of the RAD51-ATP filament based exclusively on one type of interface, we find that the pitch lengths of these model filaments are remarkably similar to the values obtained from single-molecule measurements with optical tweezers.
- Combining these experimental results, we conclude that RAD51 monomers in RAD51-ATP-ssDNA filaments can adopt two different conformations, an extended and a more compact form. These can easily interconvert due to the relatively small energy difference between them (4 kBT). The immediate implication of our findings is that the observed flexibility in the RAD51-ATP filament structure could be crucial during the search for homology, to facilitate repeated events of micro-homology sampling of heterologous dsDNA and to facilitate the subsequent strand-exchange process.

Our last point is directly relevant to the biological significance of our findings. While our studies are focused on the biophysical properties of the RAD51-ATP-ssDNA filament, we believe that they are key to understand its function, as the nucleoprotein structure that is responsible for homology search and strand exchange during homologous recombination. Several earlier studies, mentioned in our Introduction, had provided static snapshots of the Rad51 filament, providing evidence of variations in pitch. In our paper, we combine single-molecule and structural approaches to demonstrate that the RAD51-ATP-ssDNA filament exists in fact in two specific, interconvertible states. The ensuing conformational plasticity provides an appealing structural basis for the reactions of homology recognition and strand exchange that underpin homologous recombination.

We trust that these considerations, which we can readily incorporate in a revised manuscript, together with the attached reply to the reviewers' comments, will persuade you that our manuscript merits publication in The EMBO journal.

Tentative point-to-point reply to the reviewers of “Two distinct conformational states define the interaction of human RAD51 - ATP with single - stranded DNA”

Referee #1

(Report for Author)

The authors show that ATP-bound human (hRAD51) can exist in two different conformational states via a combination of single-molecule optical trapping and x-ray crystallography. This study is technically well executed and provides an intriguing insight into RAD51-ssDNA filament dynamics, albeit with limited discussion on the implications for in vivo recombination. Another strength of this study is that the single-molecule biophysics is married with a new crystal structure of the hRAD51 filament (in the absence of DNA). While overall sound, the analysis had relatively low statistics (~4 molecules) for some of the conclusions. I could also not follow a few technical details and several figures were mislabeled. In sum, I think this manuscript reports a significant finding that will be broadly interesting to readers of this journal. Below, I include suggestions that will improve clarity and possibly increase the biological significance:

Main points:

- 1. Is RAD51 disassembly sequence dependent? Some puncta seem to dissociate much faster than others in Fig 1 & 2. The high AT-content in half of their DNA substrate could let the authors analyze whether disassembly is faster/slower there.*

We have not analyzed the sequence-dependence of RAD51 disassembly. Indeed, as the referee mentions, our DNA substrate is AT-rich in one half and GC-rich in the other half, which in principle gives us the opportunity to investigate sequence effects. However, it is currently not possible to know the orientation of the DNA substrate in our optical trap, so we do not know which half of the template is actually GC-rich and which half is AT-rich. However, we will analyze the dissociation rate of both halves of each individual RAD51-ssDNA complex separately, and can determine whether there is a significant difference between these rates, giving an indication of the sequence-dependence of the dissociation rate.

- 2. Does figure 1E/2B/3A-C show fits to a single ssDNA-RAD51 filament? An average of at least a half dozen molecules with appropriate estimates of the uncertainty from multiple replicates (S.D. or 95% C.I.) will be more useful.*

The figures that are listed here indeed show typical examples, so single ssDNA molecules, each bound by multiple RAD51 filaments. However, as we will indicate more clearly in the text, the rates we report are always averages over multiple molecules (the number of molecules is stated in the text/figure captions).

- 3. I couldn't follow the logic for the 4 kBT measurement reported in relation to Figure 3. Please clarify how this number was derived. This is an important point that should be included in the supplement.*

We will include an in-depth derivation in the supplementary information of our approach to obtain this 4 k_BT free energy difference between the two conformational states.

4. *The authors assume that the persistence length and stretch modulus of the filament do not change between the two states. This assumption should be rationalized more carefully.*

We agree that this deserves more explanation than we have provided. We will add an additional discussion of this issue in the revised version of the manuscript. It is important to realize that the coverage of the ssDNA is only about 80%. We have evidence that the RAD51-ATP-ssDNA filaments are, under the current conditions, relatively short, as the filament elongation rate is comparable to nucleation rate. We know that RAD51 filaments on dsDNA are very stiff (van Mameren et al. Biophysical Journal, 2006). Most likely, the flexibility of our RAD51-ATP-covered ssDNA mostly comes from the naked stretches of ssDNA between the small filaments. In such a situation, the worm-like chain model can be used as an approximation to describe the mechanical properties of such a structure, in terms of persistence length and contour length. Of course, a complete quantitative analysis of the mechanics of extended and compact conformations would require independent fits of filaments completely in the extended conformation and completely in the compact conformation. This data, is however only available for the extended conformation, from the relaxation force-extension curve. The extension curve cannot be used for fitting, since the ratio extended / compact changes along the curve. As an approximation, we assume that it is only the contour length that changes and not the persistence length. Two aspects support the choice for this approximation. First of all, the persistence length is short (2.5 nm). Even substantial changes in this (short) persistence length would only result in subtle changes of the force-extension curves. In other words, our estimates are only weakly dependent on this parameter. Second, if we make the assumption that the persistence length is not different between the two conformations, we obtain contour length versus DNA extension curves that are close to linear (red curve in fig. 3F, including inset), as would be expected. From these curves, we obtain lengths of RAD51 monomers in extended or compacted form that are completely reasonable. It is important to realize that this quantification is an estimate that strongly depends on parameters such as percentage coverage (even more so than the exact persistence length).

5. *Include a table of relevant crystallographic information related to Figure 4.*

This table is included now (unfortunately it was not included in the original submission, but later corrected).

6. *In supplementary figure 3D, the data shown with error bars seem that there is no significant difference between ADP and ATP on the disassembly. Clearly mention how many molecules were measured, how statistical test was performed to conclude their argument.*

The number of molecules that are included in this analysis (16 for the ATP condition and 11 for the ADP condition) will be stated directly in the caption of figure 3D. We conclude that there is no significant difference between these conditions from the fact that the error bars (representing S.E.M.) overlap with each other.

7. *The difference between hRAD51-ssDNA filaments and bare ssDNA was difficult to see in figure 3A. Please clarify.*

For the case of bare ssDNA, the extension and relaxation curves completely overlap (and are distinct from the hRAD51-ssDNA curves). We will explain this in more detail in the caption of the figure.

8. *Figure S3 shows that ADP-bound RAD51 is more stable than ATP-bound RAD51. This is in direct disagreement with the reference cited (Ref. 19) that the ADP-bound is an intermediate to disassembly. This needs to be clarified.*

The main point of figure S3 is that there is no difference in stability between ADP-bound RAD51 and ATP-bound RAD51, in accordance with Ref. 19. We now realize that the caption of the figure might be misleading. We will change the caption in order to bring across this message more clearly. In agreement with Ref. 19, we propose in our model (figure 6) that ADP-bound RAD51 is indeed an intermediate to disassembly, i.e. ATP hydrolysis has to occur before disassembly takes place.

9. *The goodness of fit on figure S3C needs to be clarified.*

In the revised version of the manuscript, we will include a measure of the goodness of fit.

10. *I believe that the captions in Figs S5 and S6 actually relate to Fig 3, not Fig 4 (as written).*

Indeed, the captions refer to the wrong figure. We apologize for this oversight and will correct this in the revised version of the manuscript.

11. *The authors should speculate more about the biological significance of the two states. Do these possibly participate in heteroduplex rejection? BRC-repeat interactions?*

Our study reveals that the catalytic competent ATP-coordinated RAD51/ssDNA nucleoprotein filament (NF) is a highly flexible entity that can readily (quasi-spontaneously) switch between different conformations with different lengths. We will speculate and elaborate on the mechanistic implications that this property might have on its activities. We will center the discussion along the following lines:

(i) A plastic and flexible NF would in principle be advantageous for the homology recognition process, as it would confer conformational freedom during interrogation of duplex DNA by micro-homology sampling. We will discuss this point in relation to the studies of Eric Greene on homology recognition by RAD51 NFs. In addition, we will discuss our finding in relation to the studies of Simon Boulton on the worm Rad51 NFs where he found that maintaining a flexible pre-synaptic NF is important for homologous recombination, reinforcing the idea that a plastic NF is biologically significant.

(ii) An important aspect of the homology search process is the requisite for reversibility, as subsequent strand exchange should engage only when sufficient and correct homology is found but disengage otherwise to avoid the risk of unwanted DNA rearrangements. Flexibility of RAD51 NFs would favor reversibility.

(iii) The quasi-spontaneous interconversion between two conformations raises the possibility that the RAD51 NF could move or slide along DNA. Work from the Ha lab has revealed that sliding of bacterial RecA ortholog NFs along a duplex DNA tract facilitates the homology search process. There is, however, no published evidence for sliding by RAD51 NFs.

(iv) While the process of strand exchange remains poorly understood at the molecular level, it seems reasonable to postulate that a plastic NF would be better suited to perform such a complex three DNA strand gymnastic.

(v) It is possible that NF flexibility may contribute to facilitate BRCA2-dependent pre-synaptic NF assembly at ds/ssDNA transitions.

(vi) Finally, our new crystal structures will be very useful for modeling and molecular dynamics studies aimed at understanding the conformation of ssDNA within the NF and possibly during strand exchange (see Mara Prentis work).

12. *The authors use a new RAD51 purification protocol. This should be explained in the methods*

A manuscript describing the expression and purification of recombinant human RAD51 used for crystallization and X-ray diffraction analysis is in preparation and will be submitted shortly.

Briefly, the protocol consists in the bacterial co-expression of full-length, human RAD51 with the BRCA2 BRC4 sequence fused to a dual His-MBP tag. After initial purification of the His-MBP-BRC4 – RAD51 complex by Ni²⁺-NTA chromatography, RAD51 is separated by the BRC4 fusion protein using heparin Sepharose chromatography. This method allows for the rapid and efficient recovery of milligram amounts of pure, human RAD51 from one liter of BL21(DE3) Rosetta cells. This short description will be included in the manuscript.

Minor points:

1. *In Figures 1 - 3, quantification of more molecules with appropriate statistical tests with reinforce conclusions.*

Please note that the data we present is obtained one molecule at a time; experiments are technically extremely challenging and time-consuming. As a consequence, it is not readily feasible to obtain data on more molecules. Key problem is that DNA molecules tend to break prematurely, ending our experiments. In only a small fraction of experiments, we can obtain a complete data set. However, in these incomplete data sets we have obtained very similar behavior. Moreover, in line with major point #2 of referee #1, we will indicate more clearly in the text and figure captions how many molecules are analyzed in each experiment.

2. *Multiple figure panels frequently change axis dimensions although displaying similar data (Ex. Figure 3E/F).*

We will go carefully through the manuscript and make the axes similar where applicable.

3. *Add letters used in Figure 5A to Figure 4 to show the different RAD51 subunits in the crystal structure.*

This change will be implemented in the revised version of the manuscript.

4. *The text mentions that "there was significant hysteresis between the curves", but this sounds strange because the two curves together means hysteresis. It would be more appropriate to say "the force-extension curve shows hysteresis".*

We will reformulate this in the revised version of the manuscript.

Referee #2

(Report for Author)

Review on Brouwer et al.

EMBO-2017-98162

The paper by Brouwer et al. describes single-molecule analysis of human RAD51 nucleoprotein filaments and crystal structure analysis of human RAD51 filament. RAD51, a homolog of bacterial RecA, is essential for homologous recombination in eukaryotes. RAD51 is bound to single-stranded (ss) DNAs to form nucleoprotein filament (NPF). In the presence of ATP, RAD51 NPF is active for homology search and strand exchange with homologous double-stranded (ds) DNAs. To know the molecular mechanism of homology search in homologous recombination, it is important to decipher the dynamics of RAD51-NPF bound to ATP. The paper consists of three inter-related studies of RAD51-NPF. First, the authors analyzed dissociation dynamics of single RAD51 NPF and found that dissociation of RAD51 from ssDNAs is independent of tension of ssDNA. This is different from those of RAD51 from dsDNAs, which is dependent of the tension (as shown by one of the authors' group previously). Second, by applying forces for extension and contraction on the RAD51-NPF, the authors revealed a novel physical property of ATP-bound RAD51-NPF (not ADP-bound form), which shows "hysteresis", suggesting the presence of two different physical states of RAD51 protomers in the NPF, which are likely to be extended and contracted forms. The transition might be unidirectional, once becomes the extended form, the extended form does not become a contracted form. Third, they also determined a crystal structure of human RAD51-ATP filaments (without any DNAs in it) and found that two interfaces of RAD51 protomers. This is consistent with a previous report of yeast Rad51 filament. These results are very interesting. Particularly hysteresis in ATP-bound RAD51 NPF is very much novel physical property which would be seen in other protein machinery. The results in the paper might be suitable for publication in EMBO journal. However, there are some concerns, which should be addressed prior to publication.

- 1. One major concern is that interpretation of the connection of two results; hysteresis properties obtained by single-molecule analysis and two interfaces of RAD51 monomers. To support this, the authors need more analysis. For me, most simple (maybe difficult) one is to determine the crystal structure of RAD51 filament in the presence of "ADP". Since ADP-bound RAD51 filament does not show hysteresis (Fig. 3G), the authors would expect single interface in ADP-bound RAD51 filament. If this kind of result is provided, although this is still just additional correlation, the authors may strengthen the main conclusion.*

We agree that obtaining an ADP structure would be very interesting. However, this constitutes a study in itself and is definitely not feasible within a reasonable amount of time. We do agree that we could explain our results and in particular the connection between the two conformational states we observe both in our single-molecule experiments (as evidenced by the hysteresis) and the crystal structure (showing two clearly different RAD51 conformations) much more clearly. We will include a discussion along the following lines:

- Our single-molecule force-extension data shows clear hysteresis between extension and relaxation curves. This hysteresis can only be understood in terms of force-induced conformational changes of the RAD51-ssDNA nucleoprotein filament that result in changes of length of the filament. From our force extension curves, we can estimate the length difference of the two conformations and the mechanical work required to go from compact to extended conformations.
- Our crystal structures show that RAD51 can be in either of two conformations in a single filament, showing, amongst others different protein-protein surfaces in the filament. We can construct two models of RAD51 monomers that are purely in either conformation and find that the lengths of these model filaments are very similar to the values obtained using optical tweezers.
- From these two experimental results, we can come to a picture of RAD51-ATP-ssDNA filaments in which the RAD51 monomers can organize in two conformations, an extended and a more compact form, with relatively small energy difference between them 4 $k_B T$. In our view, this results in filaments that are highly flexible in length, which could be crucial in homology search to facilitate micro-homology sampling in double-stranded DNA, to confer reversibility and perhaps also to facilitate the subsequent strand exchange process.

Minor points:

1. *Page number should be provided.*

Page numbers will be added in the revised version of the manuscript.

2. *Figure 1: The authors should analyze the variations of dissociation of RAD51 at a local level in a way to address whether cooperative or processive dissociation of RAD51 filaments from the end occurs or not.*

Dissociation from filament ends in a pause-burst mechanism has been studied extensively for RAD51 dissociation from double-stranded DNA (van Mameren et al., Nature 2009). In that published study, we showed that ATP hydrolysis of the RAD51 monomers at filament ends is critical for dissociation to occur. In our current manuscript, we show that on ssDNA, as for dsDNA, dissociation of RAD51 depends critically on ATP hydrolysis. We also show that, in contrast to detachment from dsDNA (which is strongly DNA-tension dependent), detachment from ssDNA is mostly insensitive to tension. In kymographs of RAD51 detaching from ssDNA (such as shown in figure 1D and 2A), we observe a similar stepwise reduction of fluorescence signal of the individual filaments as on dsDNA. From this, we infer that the dissociation of RAD51 from ssDNA occurs through the same pause-burst mechanism as on dsDNA. Since we did not consider this to be the key element of our manuscript (which focusses on the two conformational states of RAD51 with ATP bound), we have not elaborated upon this. In experiments with RAD51 on ssDNA, however, we observed this peculiar hysteresis effect in the force-extension curves, which we focused the rest of our manuscript on.

Most likely, a proper analysis of the dissociation of RAD51 from ssDNA along the lines suggested by the reviewer will require additional experiments. If reviewer and editor deem this an important point, we are happy to include a more complete analysis.

3. *To make general readers understand hysteresis, it would be better to write the property of ssDNA alone (in a separate figure from Fig. 3A), which shows no hysteresis. Then, the authors should compare ATP-bound and ADP-bound RAD51-ssDNA in terms of hysteresis (Fig. 3A versus 3G) prior to indicating the hysteresis properties of ATP-bound RAD51-ssDNAs.*

From the comment of the reviewer we now understand that the concept of hysteresis and how it leads to the interpretation of two conformational states of RAD51 needs to be explained in much more detail. We will include this extended explanation in the revised version of the manuscript.

4. *For structure determination of RAD51-ATP filaments, how the authors add ATP in the crystal. In method, at least in hanging-drop buffer, no description of ATP. Do they soak RAD51-crystal with ATP?*

Human RAD51 was crystallized in the presence of ATP. For this, 2mM MgATP was added to the protein samples shortly before crystallization.

5. *Figure 5A and legend: There is clear gap between the figure and the legend. Backbone ribbons are pale blue and red, but in the legend, green and red. I can not understand what A-H means in the bottom of Figures. The legend said different chains, but no 7 blue chains init.*

The A-G and H-N annotation refers to 14 (2x7) chains present in the asymmetric unit of the Rad51-ATP crystals. The annotation is incomplete, as it is missing the color-based indication (red or green) of the interface type. We apologize for this oversight, which will be corrected in the revised version of the manuscript.

Referee #3

(Report for Author)

This manuscript combines two different biophysical characterizations of human Rad51: single-molecule force-extension studies of Rad51 - ssDNA filaments, and a crystal structure in the presence of ATP.

1. *While the crystal structure is a truly impressive feat, the overall message of this work is confusing. The force-extension curves are interpreted to reflect two differently-extended filaments for the hRad51-complex, and described as transitioning cooperatively between them. The crystal structure does show two slightly different protein-protein interfaces in the filaments, alternating. That they alternate is very intriguingly similar to the yeast Rad51 crystal structure, but it doesn't seem to connect well with the in vitro data.*

We believe that there is a very strong connection between the two datasets, but we realize now that it requires much more explanation than we have provided in the current manuscript. We will revise the manuscript to explain this much better, following the line of reasoning outlined in our answer to the major comment of referee #2.

2. *The interpretation of the single-molecule data is confusing. Could the hysteresis simply reflect a tendency of short hRad51-ATP filament segments to interact with one another (or with free ssDNA segments, in a pseudo strand-pairing reaction)? Why is the shorter filament derived modelled from the crystal structure assigned to one of the single-molecule-determined types when its length is really within error of both (1.69 uM per protomer from the crystal vs. 1.4 +/-*

0.03 and 2.2 +/- 0.6)? Why is it reasonable to assume, when determining these lengths from the single molecule data, that the persistence length of the two types would be the same?

In our single-molecule setup, the DNA molecules are held by the two ends via polystyrene microspheres. In the force-extension experiments used to study the hysteresis, the DNA ends are held at a minimum distance of around 11 μm . At this distance, the DNA is extended, and individual (RAD51-bound) segments of ssDNA are not able to interact with one another.

Therefore, the suggested interpretation of the referee (different RAD51 filaments interacting with each other or with free ssDNA segments) is not possible.

The remarks of this referee on our assumption that the persistence length of the two forms of the filaments are the same is identical to a point raised by reviewer #1 (major point 4). We refer to our answer there for further details, but we agree that this point deserves more explanation than given in the current manuscript and we will elaborate upon this in the revised version.

3. *Why is figure S3 titled "disassembly ... depends on the presence of ADP or ATP ..." when no nucleotide-free data is shown, and the two rates determined in this figure seem to be the same within error (and are referred to as the same at some point in the main text)?*

We now realize the title of this figure is very confusing, and we will change it in the revised version of the manuscript.

4. *Given that disassembly rates for the ADP- and ATP-bound states appear to be similar, why does figure 6 and its discussion assume that ATP hydrolysis must precede disassembly?*

Our data (figure 3B) as well as a previous work on double-stranded DNA (van Mameren et al., Nature 2009) shows that disassembly of RAD51 from DNA does not take place in conditions that do not allow ATP hydrolysis (replacing Mg^{2+} with Ca^{2+} in the buffer conditions). Therefore, ATP-hydrolysis is critical to allow for disassembly to occur. However, since the rates of disassembly are similar for ADP- and ATP-bound RAD51, ATP hydrolysis does not seem to be rate-limiting in RAD51 disassembly. We will state this more clearly in the text of the revised manuscript.

5. *What is the biological significance of a pathway for hRad51-ssDNA filament disassembly?*

We will discuss this point as follows:

Our study shows that RAD51 ssDNA NFs intrinsically dissociate, indicating that other factors are required for stabilization. There are a number of accessory proteins that have been implicated in NF stabilization including for instance the RAD51 paralogs. Stabilization of RAD51 ssDNA NFs combined with increased flexibility would be important for strand exchange during homologous recombination (see Simon Boulton's recent publications on this) but potentially also for its other function, which is to "protect" ssDNA in perturbed/stalled DNA replication forks. In the latter case, NFs disassembly would be required after replication restart to avoid engaging into unwanted strand exchange transactions.

6. *Figure 6 includes a confusing reference to BRC4 that needs a bit more context.*

We will remove this reference to BRC4 in this figure caption.

7. *It would help the reader to include a summary chart listing all the different rates and lengths determined in the various assays.*

Such a table will be included in the revised version of the manuscript.

8. Finally, the crystal structure, while it appears to be carefully refined, needs a bit more documentation:
- a. *The protein purification method is described simply as "unpublished". Given recent pushes for reproducibility in publications, the full methodology for reproducing this protein sample should be described.*
Please see reply to Point 12 of Referee 1.
 - b. *No mention of ATP is given in the methods - it just shows up in the picture. Please describe how much ATP was added, whether Mg⁺⁺ or Ca⁺⁺ were present, and be explicit about how it was kept from hydrolyzing during crystallization.*
RAD51 was crystallized in the presence of 2mM MgATP. The ATPase activity of human RAD51 is dependent on ssDNA, and it is negligible in its absence.
 - c. *The molecular replacement problem (orienting 14 protomers with a partial search model at ~4Å resolution) sounds truly daunting, and it would be nice for others with tough problems if a more were said about how it was tackled.*
Molecular replacement was performed using the ATPase domain of human RAD51 as search model (PDB id 1N0W). We searched in PHASER for multiple solutions and scored solutions as successful when two ATPase domains were juxtaposed by PHASER in a manner that was in agreement with the known interaction mode of two RAD51 protomers in a filament. The successful solutions were then kept as fixed solution in a successive round of MR. By iteration of the process, we were able to obtain the positions of all 14 chains in the asymmetric unit. We will add a more detailed explanation of how we solved the Molecular Replacement problem in the Methods.
 - d. *Only a 2Fo-Fc (weighted?) is shown. Due to the strong possibilities of model bias at such low resolution and with no experimental phases, it would be nice to see some minimally-biased omit maps.*
We agree with the reviewer about the potential presence of model bias in MR maps and we will add the results of the omit-map analysis to the revised manuscript. We would like to add that important features of the structure, such as the N-terminal domain of RAD51 and the ATP molecule, which were not part of the original search model, were clearly visible in the map of the successful MR solution.
 - e. *Are the details of the two interfaces seen in this work similar to those seen for yeast Rad51, or just the concept of two alternating interfaces?*
The relative orientation of the protomers in the two interface types are rather different to what was observed in the yeast RAD51 structure. We believe that the difference might be due to the absence of a bound nucleotide in the yeast RAD51 structure (although ATP γ S was used in the crystallization), as the pivot point for the composite rotation between juxtaposed RAD51 protomers is at the ATP binding pocket.
 - f. *(just a typo) in the methods section "... model for human ATP" should be "... the human Rad51-ATP complex"*
This will now be fixed in the revised manuscript.

Thank you for response letter and proposal for revising your manuscript in response to the comments of our three referees. After some delay (due to travel and absence from the office), for which I would like to apologize, I have now had a chance to carefully consider your responses. In conclusion, I overall agree to your revision plans, and we shall therefore be happy to consider an accordingly revised manuscript further for publication. In this respect, I understand that obtaining additional RAD51 crystal structures in the presence of ADP would be beyond the scope of this revision, and I also agree that in-depth analyses of RAD51 dissociation from ssDNA (as asked by referee 2) beyond the discussion provided in your response letter would also not be immediately pertinent to the main message of the paper. On the other hand, it will clearly be important to provide sufficiently detailed descriptions of the crystallization methods, pending publication of the separated manuscript dedicated to fully describing this.

I should add that it is our policy to allow only a single round of revision, making it important to carefully revise and answer all points raised to the referees' satisfaction at this point. Furthermore, please note that competing manuscript appearing elsewhere during the revision period will not negatively affect our final decision on your study. Additional information and more detailed guidelines on how to prepare a revision can be found below and in our online Guide to Authors.

Thank you again for the opportunity to consider this work! I look forward to your revision.

Point-to-point reply to the reviewers of “Two distinct conformational states define the interaction of human RAD51 - ATP with single - stranded DNA”

Referee #1

(Report for Author)

The authors show that ATP-bound human (hRAD51) can exist in two different conformational states via a combination of single-molecule optical trapping and x-ray crystallography. This study is technically well executed and provides an intriguing insight into RAD51-ssDNA filament dynamics, albeit with limited discussion on the implications for in vivo recombination. Another strength of this study is that the single-molecule biophysics is married with a new crystal structure of the hRAD51 filament (in the absence of DNA). While overall sound, the analysis had relatively low statistics (~4 molecules) for some of the conclusions. I could also not follow a few technical details and several figures were mislabeled. In sum, I think this manuscript reports a significant finding that will be broadly interesting to readers of this journal. Below, I include suggestions that will improve clarity and possibly increase the biological significance:

Main points:

- 1. Is RAD51 disassembly sequence dependent? Some puncta seem to dissociate much faster than others in Fig 1 & 2. The high AT-content in half of their DNA substrate could let the authors analyze whether disassembly is faster/slower there.*

In the optical trapping assay used, the orientation of the DNA molecule under tension is unknown. Therefore, even though our DNA substrate is AT-rich in one half and GC-rich in the other half, we do not know which half is which. However, we have now analyzed the dissociation rates on both halves of each individual RAD51-ssDNA complex separately (Appendix Figure S3). We show that there is no significant difference between the two halves, suggesting that there is no strong sequence dependence on the RAD51-ssDNA disassembly process.

- 2. Does figure 1E/2B/3A-C show fits to a single ssDNA-RAD51 filament? An average of at least a half dozen molecules with appropriate estimates of the uncertainty from multiple replicates (S.D. or 95% C.I.) will be more useful.*

The figures that are listed here indeed show typical examples of individual ssDNA molecules, each bound by multiple RAD51 filaments. However, as we have now indicated more clearly in the text, the rates we report are always averages over multiple molecules (the number of molecules is stated in the text/figure captions).

3. *I couldn't follow the logic for the 4 kBT measurement reported in relation to Figure 3. Please clarify how this number was derived. This is an important point that should be included in the supplement.*

We have now included an in-depth derivation in the appendix of our approach to estimate the 4 k_BT free energy difference between the two conformational states.

4. *The authors assume that the persistence length and stretch modulus of the filament do not change between the two states. This assumption should be rationalized more carefully.*

We agree that this deserves more explanation than we provided in the original manuscript. We have therefore added a discussion of this issue in the revised version of the manuscript.

Include a table of relevant crystallographic information related to Figure 4.

This table is now included as Appendix Table 1.

5. *In supplementary figure 3D, the data shown with error bars seem that there is no significant difference between ADP and ATP on the disassembly. Clearly mention how many molecules were measured, how statistical test was performed to conclude their argument.*

The number of molecules that are included in this analysis (16 for the ATP condition and 11 for the ADP condition) is now stated directly in the caption of the figure (now Extended Data Figure 1). We conclude that there is no significant difference between these conditions from the fact that the error bars (representing S.E.M.) overlap with each other. This is now mentioned in the caption of this figure.

6. *The difference between hRAD51-ssDNA filaments and bare ssDNA was difficult to see in figure 3A. Please clarify.*

For the case of bare ssDNA, the extension and relaxation curves completely overlap (and are distinct from the hRAD51-ssDNA curves). This is now explained in the caption of the figure.

7. *Figure S3 shows that ADP-bound RAD51 is more stable than ATP-bound RAD51. This is in direct disagreement with the reference cited (Ref. 19) that the ADP-bound is an intermediate to disassembly. This needs to be clarified.*

The main point of the figure (now Extended Data Figure 1) is that there is no difference in stability between ADP-bound RAD51 and ATP-bound RAD51, in accordance with Ref. 19. We now realize that the caption of the figure might be misleading. We have changed the caption in order to convey this message more clearly. In agreement with Ref. 19, we propose in our model (figure 6) that ADP-bound RAD51 is indeed an intermediate to disassembly, i.e. ATP hydrolysis has to occur before disassembly takes place.

8. *The goodness of fit on figure S3C needs to be clarified.*

In the revised version of the manuscript, we have included the Pearson's correlation coefficient as a measure of the goodness of fit.

9. *I believe that the captions in Figs S5 and S6 actually relate to Fig 3, not Fig 4 (as written).*

Indeed, the captions referred to the wrong figure. We apologize for this oversight and have corrected this in the revised version of the manuscript.

10. *The authors should speculate more about the biological significance of the two states. Do these possibly participate in heteroduplex rejection? BRC-repeat interactions?*

Our study reveals that the catalytically competent ATP-bound RAD51-ssDNA nucleoprotein filament (NPF) is a highly flexible entity that can readily switch between two distinct conformations with different lengths. As requested by the reviewer, we have revised and elaborated the discussion about the mechanistic implications of our findings for the reaction of strand-exchange.

11. *The authors use a new RAD51 purification protocol. This should be explained in the methods*

We have added a concise description of our protocol for Rad51 purification in the Methods. A full description of the purification protocol will be submitted to the bioRxiv preprint server shortly.

Minor points:

1. *In Figures 1 - 3, quantification of more molecules with appropriate statistical tests with reinforce conclusions.*

Please note that the data we present is obtained one molecule at a time; experiments are technically extremely challenging and time-consuming. As a consequence, it is not readily feasible to obtain data on a large number of molecules. A key problem is that DNA molecules tend to break prematurely, ending our experiments. In only a small fraction of experiments, we can obtain a complete data set. However, in these incomplete data sets we have obtained very similar behavior. Moreover, in line with major point #2 of referee #1, we have now indicated more clearly in the text and figure captions how many molecules are analyzed in each experiment.

2. *Multiple figure panels frequently change axis dimensions although displaying similar data (Ex. Figure 3E/F).*

We have gone carefully through the manuscript and standardized the axes' dimensions, where applicable.

3. *Add letters used in Figure 5A to Figure 4 to show the different RAD51 subunits in the crystal structure.*

We annotated the RAD51 subunits in Figure 4B using the letter scheme of Figure 5A, as requested by the reviewer.

4. *The text mentions that "there was significant hysteresis between the curves", but this sounds strange because the two curves together means hysteresis. It would be more appropriate to say "the force-extension curve shows hysteresis".*

In accordance with the reviewer's wish, we have reformulated the appropriate sentence in the revised version of the manuscript.

Referee #2

(Report for Author)

Review on Brouwer et al.

EMBO-2017-98162

The paper by Brouwer et al. describes single-molecule analysis of human RAD51 nucleoprotein filaments and crystal structure analysis of human RAD51 filament. RAD51, a homolog of bacterial RecA, is essential for homologous recombination in eukaryotes. RAD51 is bound to single-stranded (ss) DNAs to form nucleoprotein filament (NPF). In the presence of ATP, RAD51 NPF is active for homology search and strand exchange with homologous double-stranded (ds) DNAs. To know the molecular mechanism of homology search in homologous recombination, it is important to decipher the dynamics of RAD51-NPF bound to ATP. The paper consists of three inter-related studies of RAD51-NPF. First, the authors analyzed dissociation dynamics of single RAD51 NPF and found that dissociation of RAD51 from ssDNAs is independent of tension of ssDNA. This is different from those of RAD51 from dsDNAs, which is dependent of the tension (as shown by one of the authors' group previously). Second, by applying forces for extension and contraction on the RAD51-NPF, the authors revealed a novel physical property of ATP-bound RAD51-NPF (not ADP-bound form), which shows "hysteresis", suggesting the presence of two different physical states of RAD51 protomers in the NPF, which are likely to be extended and contracted forms. The transition might be unidirectional, once becomes the extended form, the extended form does not become a contracted form. Third, they also determined a crystal structure of human RAD51-ATP filaments (without any DNAs in it) and found that two interfaces of RAD51 protomers. This is consistent with a previous report of yeast Rad51 filament. These results are very interesting. Particularly hysteresis in ATP-bound RAD51 NPF is very much novel physical property which would be seen in other protein machinery. The results in the paper might be suitable for publication in EMBO journal. However, there are some concerns, which should be addressed prior to publication.

1. *One major concern is that interpretation of the connection of two results; hysteresis properties obtained by single-molecule analysis and two interfaces of RAD51 monomers. To support this, the authors need more analysis. For me, most simple (maybe difficult) one*

is to determine the crystal structure of RAD51 filament in the presence of "ADP". Since ADP-bound RAD51 filament does not show hysteresis (Fig. 3G), the authors would expect single interface in ADP-bound RAD51 filament. If this kind of result is provided, although this is still just additional correlation, the authors may strengthen the main conclusion.

We agree that obtaining the structure of an ADP-bound RAD51 filament would be very interesting. However, this would constitute a study in itself and is definitely not feasible within a reasonable amount of time. In our revised manuscript, we now explain and discuss more clearly the connection between the two conformational states we observe both in our single-molecule experiments (as evidenced by the hysteresis) and the crystal structure (showing two clearly different RAD51 conformations).

Minor points:

1. *Page number should be provided.*

Page numbers have now been added in the revised version of the manuscript.

2. *Figure 1: The authors should analyze the variations of dissociation of RAD51 at a local level in a way to address whether cooperative or processive dissociation of RAD51 filaments from the end occurs or not.*

In the present study we have not directly studied whether dissociation of RAD51 from ssDNA occurs from filament ends. We have now stated more clearly in the revised manuscript that this is our assumption, and have explained the basis of this assumption more in detail.

3. *To make general readers understand hysteresis, it would be better to write the property of ssDNA alone (in a separate figure from Fig. 3A), which shows no hysteresis. Then, the authors should compare ATP-bound and ADP-bound RAD51-ssDNA in terms of hysteresis (Fig. 3A versus 3G) prior to indicating the hysteresis properties of ATP-bound RAD51-ssDNAs.*

From the comment of the reviewer we now understand that the concept of hysteresis and how it leads to the interpretation of two conformational states of RAD51 needed to be explained in much more detail. We have included this extended explanation in the revised version of the manuscript.

4. *For structure determination of RAD51-ATP filaments, how the authors add ATP in the crystal. In method, at least in hanging-drop buffer, no description of ATP. Do they soak RAD51-crystal with ATP?*

Human RAD51 was crystallized in the presence of ATP. For this, 2mM MgATP was added to the protein samples shortly before crystallization. This has now been clarified in the relevant section of the Methods.

5. *Figure 5A and legend: There is clear gap between the figure and the legend. Backbone ribbons are pale blue and red, but in the legend, green and red. I can not understand what A-H means in the bottom of Figures. The legend said different chains, but no 7 blue chains init.*

We have changed the colour scheme, from red and green to blue and yellow, in the revised manuscript. A-G and H-N at the bottom of panel 5A are the protomer labels in the two heptameric chains in the asymmetric unit of the crystal structure. The schematic illustration is meant to show the distribution of the two types of protomer interface within the two heptamers of the crystal structure.

Referee #3

(Report for Author)

This manuscript combines two different biophysical characterizations of human Rad51: single-molecule force-extension studies of Rad51 - ssDNA filaments, and a crystal structure in the presence of ATP.

1. *While the crystal structure is a truly impressive feat, the overall message of this work is confusing. The force-extension curves are interpreted to reflect two differently-extended filaments for the hRad51-complex, and described as transitioning cooperatively between*

them. The crystal structure does show two slightly different protein-protein interfaces in the filaments, alternating. That they alternate is very intriguingly similar to the yeast Rad51 crystal structure, but it doesn't seem to connect well with the in vitro data.

We believe that there is a very strong connection between the two datasets, but we realize now that it requires much more explanation than we have provided in the original manuscript. We have revised the manuscript to explain this much better, in line with the major comment of referee #2.

2. *The interpretation of the single-molecule data is confusing. Could the hysteresis simply reflect a tendency of short hRad51-ATP filament segments to interact with one another (or with free ssDNA segments, in a pseudo strand-pairing reaction)? Why is the shorter filament derived modelled from the crystal structure assigned to one of the single-molecule-determined types when its length is really within error of both (1.69 μM per protomer from the crystal vs. 1.4 \pm 0.03 and 2.2 \pm 0.6)? Why is it reasonable to assume, when determining these lengths from the single molecule data, that the persistence length of the two types would be the same?*

In our single-molecule setup, the DNA molecules are held by the two ends via polystyrene microspheres. In the force-extension experiments used to study the hysteresis, the DNA ends are held at a minimum distance of around 11 μm . At this distance, the DNA is extended, and individual (RAD51-bound) segments of ssDNA are not able to interact with one another. Therefore, the suggested interpretation of the referee (different RAD51 filaments interacting with each other or with free ssDNA segments) is not possible. We have now added this explanation in the revised version of the manuscript.

The remarks of this referee on our assumption that the persistence length of the two forms of the filaments are the same is identical to a point raised by reviewer #1 (major point 4). We refer to our answer there for further details, but we agree that this point deserved more explanation than was given in the original manuscript and we have elaborated upon this in the revised version.

3. *Why is figure S3 titled "disassembly ... depends on the presence of ADP or ATP ..." when no nucleotide-free data is shown, and the two rates determined in this figure seem to be the same within error (and are referred to as the same at some point in the main text)?*

We now realize the title of this figure was confusing, and we have changed it in the revised version of the manuscript.

4. *Given that disassembly rates for the ADP- and ATP-bound states appear to be similar, why does figure 6 and its discussion assume that ATP hydrolysis must precede disassembly?*

Our data (figure 3B) as well as a previous work on double-stranded DNA (van Mameren et al., Nature 2009) shows that disassembly of RAD51 from DNA does not take place in conditions that do not allow ATP hydrolysis (replacing Mg^{2+} with Ca^{2+} in the buffer conditions). Therefore, ATP-hydrolysis is critical to allow for disassembly to occur. However, since the rates of disassembly are similar for ADP- and ATP-bound RAD51, ATP hydrolysis does not seem to be rate-limiting in RAD51 disassembly. We have now stated this more clearly in the text of the revised manuscript.

5. *What is the biological significance of a pathway for hRad51-ssDNA filament disassembly?*

Our study shows that RAD51-ssDNA NPFs are dynamic entities that dissociate spontaneously over a time scale of minutes, indicating that other protein factors might be required to promote their stabilization. Indeed, recent work from the Boulton's lab (Taylor et al, Mol Cell, 2016) has shown that the RAD51 paralog in worms, RFS-1, stabilizes Rad51-ssDNA NPFs by preventing its dissociation from the DNA via a direct conformational effect on the filament. We note that filament stabilization might also play an important role during DNA replication at stalled replication forks, when RAD51 coats the ensuing ssDNA gaps for protection from degradation. We have now added this discussion to the revised version of the manuscript.

6. *Figure 6 includes a confusing reference to BRC4 that needs a bit more context.*

We have removed the reference to BRC4 in this figure caption.

7. *It would help the reader to include a summary chart listing all the different rates and lengths determined in the various assays.*

Such a table is now included in the revised version of the manuscript (Appendix Table 2).

8. Finally, the crystal structure, while it appears to be carefully refined, needs a bit more documentation:

- a. The protein purification method is described simply as "unpublished". Given recent pushes for reproducibility in publications, the full methodology for reproducing this protein sample should be described.

We have now added a description of the purification protocol to the relevant section of the Methods. Please see reply to Point 12 of Referee 1.

No mention of ATP is given in the methods - it just shows up in the picture. Please describe how much ATP was added, whether Mg⁺⁺ or Ca⁺⁺ were present, and be explicit about how it was kept from hydrolyzing during crystallization.

Human RAD51 was crystallised in the presence of ATP. For this, 2mM MgATP was added to the protein samples shortly before crystallization. This has now been clarified in the relevant section of the Methods. The ATPase activity of human RAD51 is dependent on ssDNA, and it is negligible in its absence.

- b. The molecular replacement problem (orienting 14 protomers with a partial search model at ~4Å resolution) sounds truly daunting, and it would be nice for others with tough problems if a more were said about how it was tackled.

Molecular replacement was performed using the ATPase domain of human RAD51 as search model (PDB id 1N0W). We searched in PHASER for multiple solutions and scored solutions as successful when two ATPase domains were juxtaposed by PHASER in a manner that was in agreement with the known interaction mode of two adjacent RAD51 protomers in a filament. The successful solutions were then kept as fixed solution in a successive round of MR. By iteration of the process, we were able to obtain the positions of all 14 chains in the asymmetric unit. We have added a more detailed explanation of how we solved the Molecular Replacement problem in the Methods.

- c. Only a 2Fo-Fc (weighted?) is shown. Due to the strong possibilities of model bias at such low resolution and with no experimental phases, it would be nice to see some minimally-biased omit maps.

We agree with the reviewer about the potential presence of model bias in MR maps. We have generated an omit map for the ATP ligand using Polder in Phenix, which shows clearly the presence of positive density for the ATP ligand (Supplementary figure 7C).

We would like to add that important features of the structure, such as the N-terminal domain of RAD51 and the ATP molecule, which were not part of the original search model, were clearly visible in the map of the successful MR solution.

- d. Are the details of the two interfaces seen in this work similar to those seen for yeast Rad51, or just the concept of two alternating interfaces?

The relative orientation of the protomers in the two interface types are rather different to what was observed in the yeast RAD51 structure. We believe that the difference might be due to the absence of a bound nucleotide in the yeast RAD51 structure (although ATP γ S was used in the crystallization), as the pivot point for the composite rotation that we observe between neighbouring RAD51 protomers is at the ATP-binding pocket.

- e. (just a typo) in the methods section "... model for human ATP" should be "... the human Rad51-ATP complex"

This has now been fixed in the revised manuscript.

2nd Editorial Decision

15 January 2018

Thank you for submitting your revised manuscript, which has now been assessed once more by the original reviewers 1 and 3. Both consider the manuscript significantly improved, but as you will see, referee 3 still has several queries and concerns regarding the interpretation and presentation of the results. I feel it would be important to clarify these issues prior to publication, and would therefore like to give you an opportunity to answer to these comments by way of a final revision and accompanying point-by-point response letter.

I am therefore returning the manuscript to you for an additional round of revision, hoping that you will be readily able to satisfactorily address all remaining points. Please do not hesitate to get back to me should you have any further questions.

REFEREE REPORTS

Referee #1:

The reviewers satisfied my concerns adequately. I feel that the revised manuscript is ready for publication.

Referee #3:

The revised manuscript is much clearer but still confusing in a few aspects.

- 1) The single-molecule stretching measurements here show that the ADP form is similar to the "ATP-extended" form. (p8 states that the authors assume the lengths to be similar). But the introduction and discussion (p4 and p11) refer to the ADP state as compact. Please address the apparent direct contradiction!
- 2) I still worry that the authors' correlations between the two different states seen in the stretching data and the two interfaces in the crystal are the intellectual equivalent of jamming round pegs into square holes, which they might later regret. My reasoning is:
 - a. The newer figure 5 helps - thank you. But it shows that the two interfaces differ the least - almost not at all - at the ATP site. Much of the flexibility appears to be at the periphery, or within the individual subunits themselves. These differences might simply reflect the effects of crystal packing on a somewhat flexible molecule?
 - b. The rises of two filaments derived from the crystal structure are closer to each other than to either of the solution-derived rises, lie in-between the solution-derived rises, and with the exception of the shorter crystal filament vs. the extended solution filament, both crystal filament rises are within error of both solution filaments.
 - c. My hunch is that the ATP-extended and the ADP forms represent ones in which the subunits are still bound to their neighbors via the trans beta strand, but the interface between the two ATPase cores has become disrupted / flexible / weak. In that case, the "extended" form is really an "extensible" form. At first I thought this was incompatible with the 5-20 pN range in Figure 3a where the ATP-extended form is more extended than ssDNA, but I think these ideas might be reconciled if the ATP-extended form still binds ~3nt / subunit, but is simply less rigid about how those nucleotides relate to their neighbors? I think this "extensible / flexible" interface idea might also reconcile the rises for the ADP filaments?
- 3) That the ATP form could be crystallized with WT protein and Mg⁺⁺ is really surprising given the observation that ATP hydrolysis is fast enough that it is not rate-limiting for filament disassembly, which can be observed in real time. Does the presence of DNA make that big a difference to the ATP hydrolysis rate (if so, please give a reference)? Is the density for the every 3rd phosphate as good as that of the 1st too?
- 4) The discussion on p9 overstates the unclearness of our understanding, and should probably reference some Greene and Prentiss papers. Also, it is not true that this is the first RecA/RadA/Rad51 crystal to contain more than monomers or dimers - the Pavletich RecA-DNA structures have 10 copies in the asymmetric unit.

Minor things:

- 1) P7 - I think you mean 3 nt not 3 nm for the footprint size?
- 2) P10 - Sentence starting "ATP hydrolysis must thus precede disassembly ... does not depend on the nucleotide cofactor bound ..." is confusing. I suggest rewording to "... nucleotide cofactor initially bound ..."
- 3) I still find the cartoon in figure 6 confusing. As drawn it doesn't explain why the ATP-bound states can't directly disassemble, especially if the pulling experiments found the ATP-extended state to be similar to the ADP state. Would it help to add presumed barrier heights between states?

2nd Revision - authors' response

7 February 2018

Point-to-point reply to the reviewers of "Two distinct conformational states define the interaction of human RAD51 - ATP with single - stranded DNA"

Referee #1:

The reviewers satisfied my concerns adequately. I feel that the revised manuscript is ready for publication.

Referee #3:

The revised manuscript is much clearer but still confusing in a few aspects.

1. *The single-molecule stretching measurements here show that the ADP form is similar to the "ATP-extended" form. (p8 states that the authors assume the lengths to be similar). But the introduction and discussion (p4 and p11) refer to the ADP state as compact. Please address the apparent direct contradiction!*

Determining the length of the ADP-bound filament from our single-molecule experiments is very challenging, as the experiments with ADP-bound filaments (Figure 3G-I) are performed under conditions where hRAD51 is competent to disassemble from the ssDNA template. Therefore, the length of the ADP-state cannot be directly inferred from an eWLC fit to the relaxation curve (as was done for the ATP-extended state in Figure 3F). Conditions in the presence of ADP, where disassembly is blocked, have not been found experimentally, making it impossible to capture the ADP-state to measure its exact length. In the revised version of the manuscript, we have now moderated our conclusions about the length of the ADP-state. In accordance with previous studies, we assume that the length of the ADP-state is more compact with respect to the ATP-extended conformation.

2. *I still worry that the authors' correlations between the two different states seen in the stretching data and the two interfaces in the crystal are the intellectual equivalent of jamming round pegs into square holes, which they might later regret. My reasoning is:*
 - a. *The newer figure 5 helps - thank you. But it shows that the two interfaces differ the least - almost not at all - at the ATP site. Much of the flexibility appears to be at the periphery, or within the individual subunits themselves. These differences might simply reflect the effects of crystal packing on a somewhat flexible molecule?*

The flexibility is a property of the filament, not of the individual subunits, whose structure remains unchanged in the two filament types. That the subunit interface around the ATP-binding pocket changes relatively little is to be expected, as it represents the pivot point of the rearrangement, and it is also immaterial; the important point is that the specific change between subunits that we observe (described on page 9 as 'a rigid-body movement comprising of a slight tilt towards the filament axis coupled to a small increase in twist') gives rise to two different filament states, as shown in Figure 5 and the supplementary movies. As we argue below, it is possible that crystal packing might have affected the extent of the difference between the two interface types, but it is very unlikely to be the cause of it. We have added a sentence in the text (page 9), to reinforce the point that the structure of the individual RAD51 subunits remains unchanged in the two dimer types observed in the crystal of the filament.

- b. *The rises of two filaments derived from the crystal structure are closer to each other than to either of the solution-derived rises, lie in-between the solution-derived rises, and with the exception of the shorter crystal filament vs. the extended solution filament, both crystal filament rises are within error of both solution filaments.*

We take the point made by the reviewer concerning the absolute value of filament rises in the crystal and in solution. The reasons for the difference in rise change between filaments in solution and in the crystal can be varied, and include the fact that the ATP-RAD51 crystal structure lacked ssDNA and that it was possibly influenced by the constraint of crystal packing. Equally, the calculation for the solution measurement of filament rise are indirect and depend on a number of assumptions that, although valid, are likely to affect the absolute figure to some extent. Thus, we argue that what matters most is not the direct comparison of the actual values between solution and crystal filaments, but the striking observation that both experiments detected the presence of two different ATP-bound filament states. We have added a sentence on page 10 of the Results section, to

highlight this difference between solution and crystal measurements in the extent of the change in filament rise for the two ATP-bound states.

- c. *My hunch is that the ATP-extended and the ADP forms represent ones in which the subunits are still bound to their neighbors via the trans beta strand, but the interface between the two ATPase cores has become disrupted / flexible / weak. In that case, the "extended" form is really an "extensible" form. At first I thought this was incompatible with the 5-20 pN range in Figure 3a where the ATP-extended form is more extended than ssDNA, but I think these ideas might be reconciled if the ATP-extended form still binds ~3nt / subunit, but is simply less rigid about how those nucleotides relate to their neighbors? I think this "extensible / flexible" interface idea might also reconcile the rises for the ADP filaments?*

We are unclear as to what the reviewer envisions as the 'extensible' nature of the filament interface. Currently we don't have experimental information concerning the relative conformation of the ssDNA in the two ATP-bound forms. However, we agree with the reviewer that in both ATP- and ADP-bound forms of the filament, neighbouring RAD51 subunits remain stably associated via the short beta strand motif located between N-terminal and ATPase domains. Equally, we agree that in both ATP-bound forms, the 'footprint' of RAD51 subunit on ssDNA remains constant at 3 nucleotides, as previously observed (ref. 21).

3. *That the ATP form could be crystallized with WT protein and Mg⁺⁺ is really surprising given the observation that ATP hydrolysis is fast enough that it is not rate-limiting for filament disassembly, which can be observed in real time. Does the presence of DNA make that big a difference to the ATP hydrolysis rate (if so, please give a reference)? Is the density for the every 3rd phosphate as good as that of the 1st too?*

ATP hydrolysis by the ATP-RAD51 filament is negligible in the absence of DNA, see for instance Table 1 of Sung, *Science*, **265**, 1241-1243 (1996) for yeast RAD51; Figure 1B of Morrison et al, *MCB*, **19** 6891-6897 (1999) and Figure 5A of Wang et al., *Molecular Cell* **59**, 478-490, (2015) for human RAD51.

Within the limitation of a 3.9 Å map, the density for the triphosphate group is clear for all RAD51-RAD51 interfaces in the crystal.

4. *The discussion on p9 overstates the unclarity of our understanding, and should probably reference some Greene and Prentiss papers. Also, it is not true that this is the first RecA/RadA/Rad51 crystal to contain more than monomers or dimers - the Pavletich RecA-DNA structures have 10 copies in the asymmetric unit.*

In accordance with the request of the reviewer, we have added references to previous published work on the structure of RecA/RAD51 nucleoprotein filaments by the Greene, Pavletich and Prentiss labs, on pages 8 and 9.

Although it is true that the crystal structure of the RecA-ssDNA filament solved by Chen and colleagues contained multiple copies of the RecA protein in the asymmetric unit, these had been covalently linked in a single recombinant polypeptide. They cannot therefore be assumed to be truly conformationally independent, in the same way as the RAD51 subunits in our crystal structure.

Minor things:

1. *P7 - I think you mean 3 nt not 3 nm for the footprint size?*

The reviewer is correct, this mistake has been corrected in the revised version of the manuscript.

2. *P10 - Sentence starting "ATP hydrolysis must thus precede disassembly ... does not depend on the nucleotide cofactor bound ..." is confusing. I suggest rewording to "... nucleotide cofactor initially bound ..."*

Indeed, the sentence was somewhat confusing. We have now corrected it as suggested by the reviewer.

3. *I still find the cartoon in figure 6 confusing. As drawn it doesn't explain why the ATP-bound states can't directly disassemble, especially if the pulling experiments found the ATP-extended state to be similar to the ADP state. Would it help to add presumed barrier heights between states?*

Taking into consideration the remark of the referee, we have modified the figure, to highlight more clearly our findings and to avoid confusion. We now depict the reaction pathway from ATP-filament

to ADP filament to filament disassembly (x-axis) and include energy barriers to the measured free energy differences between the states.

3rd Editorial Decision

8 February 2018

Thank you for submitting your final revised manuscript for our consideration. I am pleased to inform you that we have now accepted it for publication in The EMBO Journal.

Corresponding Author Name: Prof. dr. E.J.G. Peterman

Journal Submitted to: The EMBO Journal

Manuscript Number: EMBOJ-2017-98162